# Immunotherapy targeting plasma ASM is protective in a mouse model of Alzheimer's disease

Byung Jo Choi[1,2,11], Min Hee Park[1,3,11], Kang Ho Park[1,3], Wan Hui Han[1,3], Hee Ji Yoon[1,3], Hye Yoon Jung[1,3], Ju Yeon Hong[1,3], Md Riad Chowdhury[1,3], Kyung Yeol Kim[1,3], Jihoon Lee[4], Im-Sook Song[4], Minyeong Pang[5], Min-Koo Choi[5], Erich Gulbins [6], Martin Reichel[7], Johannes Kornhuber [7], Chang-Won Hong [3], Changho Kim[8], Seung Hyun Kim [9], Edward H. Schuchman[10], Hee Kyung Jin [1,2] ✉ & Jae-sung Bae [1,3] ✉

Acid sphingomyelinase (ASM) has been implicated in neurodegenerative disease pathology, including Alzheimer's disease (AD). However, the specific role of plasma ASM in promoting these pathologies is poorly understood. Herein, we explore plasma ASM as a circulating factor that accelerates neuropathological features in AD by exposing young APP/PS1 mice to the blood of mice overexpressing ASM, through parabiotic surgery. Elevated plasma ASM was found to enhance several neuropathological features in the young APP/PS1 mice by mediating the differentiation of blood-derived, pathogenic Th17 cells. Antibody-based immunotherapy targeting plasma ASM showed efficient inhibition of ASM activity in the blood of APP/PS1 mice and, interestingly, led to prophylactic effects on neuropathological features by suppressing pathogenic Th17 cells. Our data reveals insights into the potential pathogenic mechanisms underlying AD and highlights ASM-targeting immunotherapy as a potential strategy for further investigation.

Extensive research has shown the importance of circulating factors in the pathogenesis of aging and age-related neurodegenerative diseases such as Alzheimer's disease (AD)[1–6]. For example, animal studies using heterochronic parabiosis, which is a technique combining the blood circulation of two animals, have revealed the powerful rejuvenating effects of young blood on aging and/or the age-related neurodegenerative brain disease[7–11]. In contrast, exposing young mice to blood from old wild type (WT) or AD mice contributed to accelerated brain dysfunction, including neuroinflammation, reduction of synaptic plasticity and neurogenesis, and impairment of cognitive function[12–16]. These findings suggest that transfusion of young blood or specific circulating factors found in this blood may be a promising therapeutic strategy for the treatment of age-related neurogenerative diseases.

[1]KNU Alzheimer's disease Research Institute, Kyungpook National University, Daegu, South Korea. [2]Department of Laboratory Animal Medicine, College of Veterinary Medicine, Kyungpook National University, Daegu, South Korea. [3]Department of Physiology, Cell and Matrix Research Institute, School of Medicine, Kyungpook National University, Daegu, South Korea. [4]BK21 FOUR Community-Based Intelligent Novel Drug Discovery Education Unit, Vessel-Organ Interaction Research Center (VOICE), College of Pharmacy and Research Institute of Pharmaceutical Sciences, Kyungpook National University, Daegu, South Korea. [5]College of Pharmacy, Dankook University, Cheon-an, South Korea. [6]Department of Molecular Biology, University of Duisburg-Essen, Essen, Germany. [7]Department of Psychiatry and Psychotherapy, Friedrich-Alexander-University of Erlangen-Nuremberg, Erlangen, Germany. [8]Department of Emergency Medicine, Kyungpook National University Chilgok Hospital, School of Medicine, Kyungpook National University, Daegu, South Korea. [9]Department of Neurology, Hanyang University College of Medicine, Seoul, South Korea. [10]Department of Genetics and Genomic Sciences, Icahn School of Medicine at Mount Sinai, New York, NY, USA. [11]These authors contributed equally: Byung Jo Choi, Min Hee Park. ✉e-mail: hkjin@knu.ac.kr; jsbae@knu.ac.kr

We previously found high activity of acid sphingomyelinase (ASM), a sphingolipid metabolizing enzyme that catalyzes the hydrolysis of sphingomyelin to ceramide, in the blood and brain of old vs. young individuals or mice[17]. It was also shown that microvessels containing blood-brain barrier endothelial cells (BBB-ECs) were the main contributor of elevated ASM activity in the old mouse brain. In addition to these findings, some researchers, including us, have observed that ASM activity was also elevated in the blood and neurons of AD mouse models or AD patients[18–21]. The pathological role of elevated ASM in the aged or AD brain was shown to be involved, at least in part, to BBB leakage, BBB-EC apoptosis, neuronal autophagy dysfunction, and/or impaired neurogenesis[17,18,21,22]. Although the role of secretory ASM, including plasma ASM, in several diseases has been described[19,23,24], little is known about the specific pathological effects of this ASM activity on age-related neurodegenerative diseases such as AD.

In this study we reveal the critical effects of elevated plasma ASM activity on neuropathological features of AD using parabiosis mouse models. Young AD mice showed accelerated amyloid beta (Aβ) deposition and neuroinflammation after exposure to blood of ASM overexpressing mice. The overexpressing ASM in the blood induced CD4[+] T cell differentiation into pathogenic Th17 cells, and contributed to the accelerated pathological changes in the young AD mouse brain. Moreover, we showed prophylactic effects of ASM inhibition on various pathological features in AD mice by sharing the circulatory system of these mice with ASM knock-out mice, as well as by using anti-ASM antibodies. Based on these findings, we suggest that antibody-based immunotherapy against ASM in the blood is a promising therapeutic strategy that prevents various pathological features in AD by regulating pathogenic Th17 cells.

## Results

### Overexpression of plasma ASM accelerates Aβ accumulation and neuroinflammation in the brain of young APP/PS1 mice

We first measured ASM activity in the plasma of AD patients and confirmed that this activity was increased as the disease progressed. Surprisingly, no changes in the levels of plasma ceramides were detected (Supplementary Fig. 1a, b). The APP/PS1 mouse, a well-studied and validated AD mouse model[21,25], also exhibited an increase of ASM activity in plasma but, as was the case in patients, no significant differences in plasma ceramides with age (Supplementary Fig. 1c, d).

Next, we investigated the contribution of elevated plasma ASM on the neuropathological features of AD. To increase plasma ASM activity, we used conditional, transgenic mice with endothelial cell-specific overexpression of ASM (Tie2-cre; $Smpd1^{ox/ox}$ mice), and joined the circulation of these mice with three-month-old WT or APP/PS1 mice by parabiotic surgery (Fig. 1a). We found that the Tie2-cre; $Smpd1^{ox/ox}$–WT and Tie2-cre; $Smpd1^{ox/ox}$–APP/PS1 parabiotic mice showed increased levels of plasma ASM activity compared to WT–WT and WT–APP/PS1 parabiotic mice at 5 weeks post-surgery (Fig. 1b). No significant differences in plasma ceramide levels were found in these mice (Fig. 1c). To determine whether the increase of plasma ASM activity was accompanied by local changes within the brain, we measured ASM activity in the cortex of these mice. Contrary to the plasma results, these was no difference in ASM activity in the cortex (Fig. 1d), indicating a specific increase of plasma ASM activity in these young WT and APP/PS1 mice that had been joined by parabiosis with ASM overexpressing mice.

Aβ accumulation is a major AD pathology that is prominent starting at ~7 months of age in the APP/PS1 mouse[25]. Notably, 3-month-old APP/PS1 mice exchanged by parabiosis with blood of Tie2-cre; $Smpd1^{ox/ox}$ mice overexpressing plasma ASM showed early Aβ accumulation by thioflavin S (ThioS) and 6E10 staining, as well as by Aβ40 and Aβ42 ELISA in the cortex (Fig. 1e–g). Additionally, these mice exhibited highly activated microglia and astrocytes (Fig. 1h, i). Although WT mice exchanged by parabiosis with blood of Tie2-cre;

$Smpd1^{ox/ox}$ mice showed slight activation of these cells in the cortex, it was less than in APP/PS1 mice exchanged by parabiosis with blood of Tie2-cre; $Smpd1^{ox/ox}$ mice. Pro-inflammatory markers, including $Tnf$-$α$, $Il$-$1β$, and $Il$-$6$ were increased in the cortex of these mice, while anti-inflammatory markers, including $Tgf$-$β$, and $Arg1$, were decreased compared to the young APP/PS1 mice exposed to WT blood (Fig. 1j). The microglia morphology of APP/PS1 mice overexpressing ASM activity exhibited several pro-inflammatory features, including increased microglial volume, cell body size, dendrite length, as the number of segments, terminal points, and branch points. Moreover, these microglia exhibited decreased phagocytic function as indicated by less lysosomal staining of Aβ in the cortex (Fig. 1k, l). We further observed early Aβ accumulation and microglia activation in the hippocampus of young APP/PS1 mice exchanged with blood of ASM overexpressing Tie2-cre; $Smpd1^{ox/ox}$ mice, despite the fact that there was no change in hippocampal ASM activity (Supplementary Fig. 2a–h). Overall, these results indicated that overexpression of plasma ASM activity contributed to acceleration of Aβ accumulation, neuroinflammation, and a decrease of microglia phagocytic function in the brain of young APP/PS1 mice despite the fact that the brain tissue ASM activity was not elevated. The early Aβ accumulation in the cortex and hippocampus of these mice was not associated with expression of APP and Aβ generating enzyme, Bace-1 (Supplementary Fig. 2i).

### Elevated plasma ASM induces apoptosis and pathogenic Th17 cell differentiation of CD4[+] T cell

It is widely accepted that blood-derived immune cells, including neutrophils, monocytes, and lymphocytes, are found in the brain of AD patients and mouse models[26,27]. In addition, changes in some leukocyte subpopulations have been associated with Aβ clearance or microglia-mediated neuroinflammation in the AD brain[28–30]. Considering that overexpressed plasma ASM led to acceleration of neuroinflammation in the brain of young APP/PS1 mice, as well as impaired phagocytic function of microglia, we hypothesized that this might be related to the effects of plasma ASM on immune cells derived from blood. Therefore, we investigated the changes of leukocyte subpopulations in WT–WT, Tie2-cre; $Smpd1^{ox/ox}$ –WT, WT–APP/PS1, and Tie2-cre; $Smpd1^{ox/ox}$–APP/PS1 parabiont mice by flow cytometry. We found that CD4[+] T cells were decreased in the blood of ASM overexpressing Tie2-cre; $Smpd1^{ox/ox}$–WT and Tie2-cre; $Smpd1^{ox/ox}$–APP/PS1 mice compared to parabionts of WT–WT and WT–APP/PS1 mice. However, the reduction of CD4[+] T cells was more markedly reduced in the blood of APP/PS1 mice exposed to Tie2-cre; $Smpd1^{ox/ox}$ compared to WT. There were no significant differences in the changes of other leukocytes subpopulations such as neutrophils, monocytes, macrophages, CD8[+] T cells, and B220[+] B cells between these groups (Supplementary Fig. 3a, b).

CD4[+] T cells could differentiate into a variety of T helper (Th) and regulatory T (Treg) cells, depending on the specific cytokine milieu[30–32]. Th1 and Th17 cells have a major role in promoting inflammation, whereas Th2 and Treg cells mainly mediate anti-inflammation and immunosuppression[30–32]. We therefore analyzed these subsets and observed a remarkable increase of Th17 cells and decrease of Treg cells in the blood of WT– WT vs. WT –Tie2-cre; $Smpd1^{ox/ox}$ mice and Tie2-cre; $Smpd1^{ox/ox}$–APP/PS1 vs. WT–APP/PS1 mice (Supplementary Fig. 3c). These changes were greater in the blood of APP/PS1 mice exchanged by parabiosis with blood of Tie2-cre; $Smpd1^{ox/ox}$ than WT mice exposed to Tie2-cre; $Smpd1^{ox/ox}$ blood. The results in the spleen and brain tissue also showed similar changes in the CD4[+] T, Th17, and Treg cells (Supplementary Fig. 3d–g). Overall, these findings suggested that elevated plasma ASM may influence survival of CD4[+] T cells and differentiation into Th17 cells in the spleen and blood of young APP/PS1 mice, and that these changes in the blood could lead to similar changes in the brain. Moreover, blood-derived Th17 cells might be involved in the microglia-mediated neuroinflammation and early Aβ accumulation we observed in young APP/PS1 mice that were overexpressing ASM.

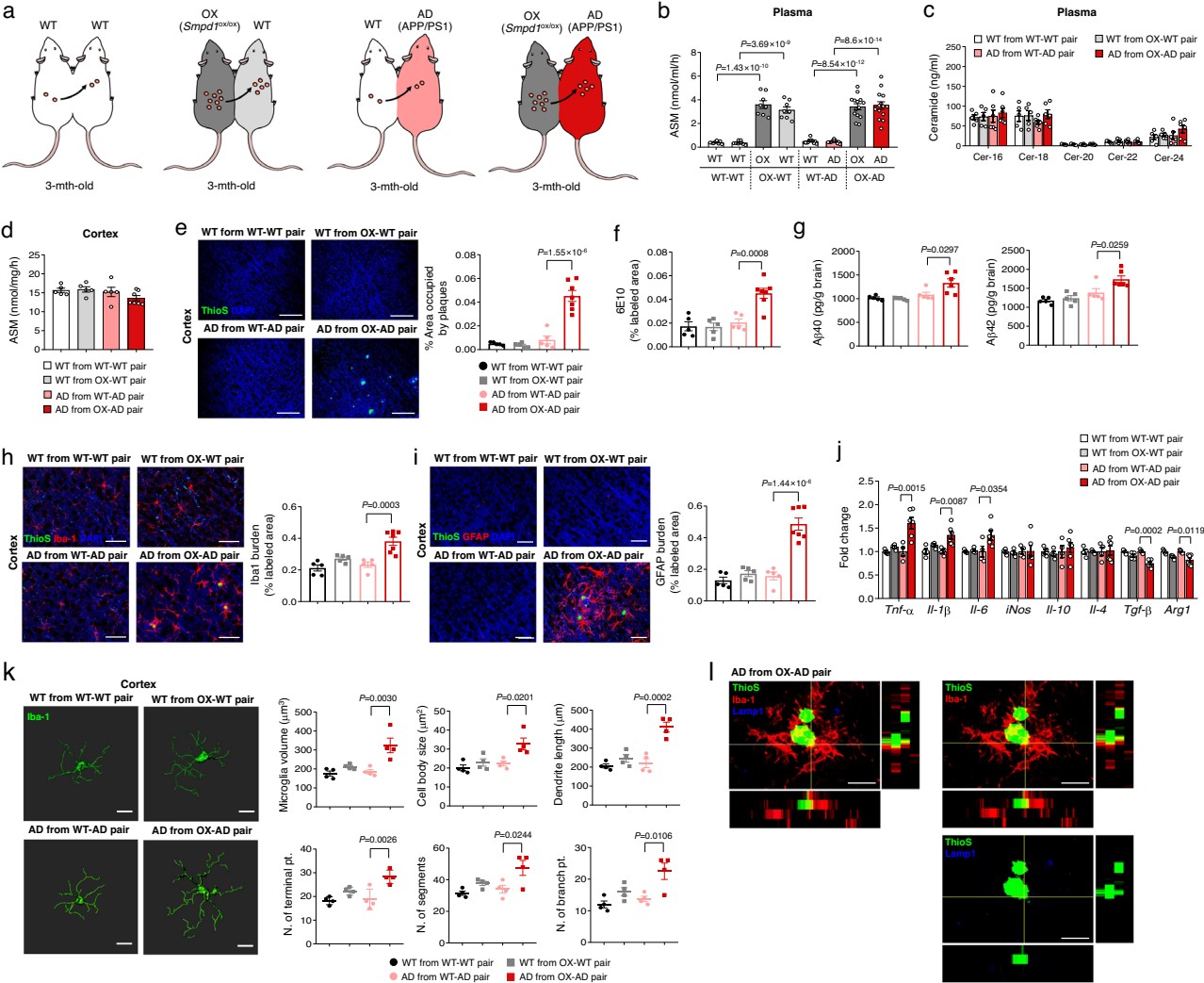

**Fig. 1 | Young APP/PS1 mice in parabiosis with *Smpd1*^ox/ox mice exhibit acceleration of Aβ deposits and neuroinflammation in the cortex. a** Schematic showing parabiotic pairings. **b** ASM activity in plasma (WT and WT from WT−WT pair, OX and WT from OX-WT pair, $n = 8$; WT and AD from WT-AD pair, $n = 12$; OX and AD from OX-AD pair, $n = 14$). **c** Ceramide concentration in plasma (WT from WT−WT pair and OX-WT pair, $n = 5$; AD from WT-AD pair and OX-AD pair, $n = 6$). **d** ASM activity in the cortex. **e** Representative immunofluorescence images and quantification of ThioS (Aβ plaques) in the cortex. Scale bars, 50 μm. **f** Quantification of 6E10 (d-f: WT from WT−WT, WT from OX-WT, AD from WT-AD pair, $n = 5$; AD from OX-AD pair, $n = 7$). **g** Aβ40 and Aβ42 depositions in the cortex (WT from WT−WT, WT from OX-WT, AD from WT-AD pair, $n = 5$; AD from OX-AD pair, $n = 6$). **h, i** Immunofluorescence images and quantification of microglia (**h**, Iba-1, red) and astrocyte (**i**, GFAP, red) with Aβ plaques (ThioS, green) in the cortex (WT from WT−WT, WT from OX-WT, AD from WT-AD pair, $n = 5$; AD from OX-AD pair, $n = 7$). Scale bars, 50 μm. **j** mRNA levels of inflammatory markers in the cortex (WT from WT−WT, WT from OX-WT, AD from WT-AD pair, $n = 4$; AD from OX-AD pair, $n = 6$). **k** Imaris-based three-dimensional images (Scale bars, 10 μm) and quantification of microglial morphology in the cortex ($n = 4$/group). **l** Immunofluorescence images of ThioS (Aβ plaques, green) encapsulated within Lamp1⁺ structures (phagolysosomes, blue) in microglia (Iba1, red) present in the cortex. Scale bars, 30 μm. AD, APP/PS1; OX, *Smpd1*^ox/ox (Tie2-cre; *Smpd1*^ox/ox mice). **b**, **e**–**k** One-way analysis of variance, Tukey's post hoc test. All error bars indicate s.e.m. All data analysis was done at 4.5-mo-old mice. Source data are provided as a Source Data file.

To further investigate the relationship of elevated plasma ASM activity on CD4⁺ T cell survival and differentiation, we performed several in vitro experiments. First, we confirmed a significant increase of ASM activity in the serum of Tie2-cre; *Smpd1*^ox/ox vs. WT mice, while those of *Smpd1*^−/− mice[33] displayed little activity (Fig. 2a). Each serum was then exposed to CD4⁺ T cells, and we observed an increase of apoptotic cells after treatment with Tie2-cre; *Smpd1*^ox/ox but not WT or *Smpd1*^−/− serum (Fig. 2b). This indicated that elevated serum ASM activity might cause apoptosis of CD4⁺ T cell in the blood. We further confirmed the effects of elevated ASM on the in vitro differentiation of Th17 or Treg cells. Stimulation in the presence of serum from Tie2-cre; *Smpd1*^ox/ox mice markedly induced IL17A expression of CD4⁺ T cells as determined by flow cytometry, while we did not observe comparable outcomes on the differentiation of Treg cells (Fig. 2c, d). Therefore,

these data suggested that elevated serum ASM activity enhanced CD4⁺ T cell differentiation into Th17 cells.

To further elucidate how ASM affects CD4⁺ T cells apoptosis and differentiation, we measured ASM activity in cell membrane and cytosol fractions of the CD4⁺ T cells after treatment with each serum. Interestingly, CD4⁺ T cells treated with the serum of Tie2-cre; *Smpd1*^ox/ox mice exhibited an elevation of ASM activity in the cell membrane fraction, but not in the cytosol, compared to WT or *Smpd*^−/− mice (Fig. 2e). The elevated ASM also led to an increase of various kind of ceramides in the cell membranes of the Tie2-cre; *Smpd1*^ox/ox mice (Fig. 2f). To confirm these in vitro observations, CD4⁺ T cells were isolated from young APP/PS1 mice that had been exchanged with blood of WT or Tie2-cre; *Smpd1*^ox/ox mice. We found an increase of ASM activity and ceramide levels in the membranes of

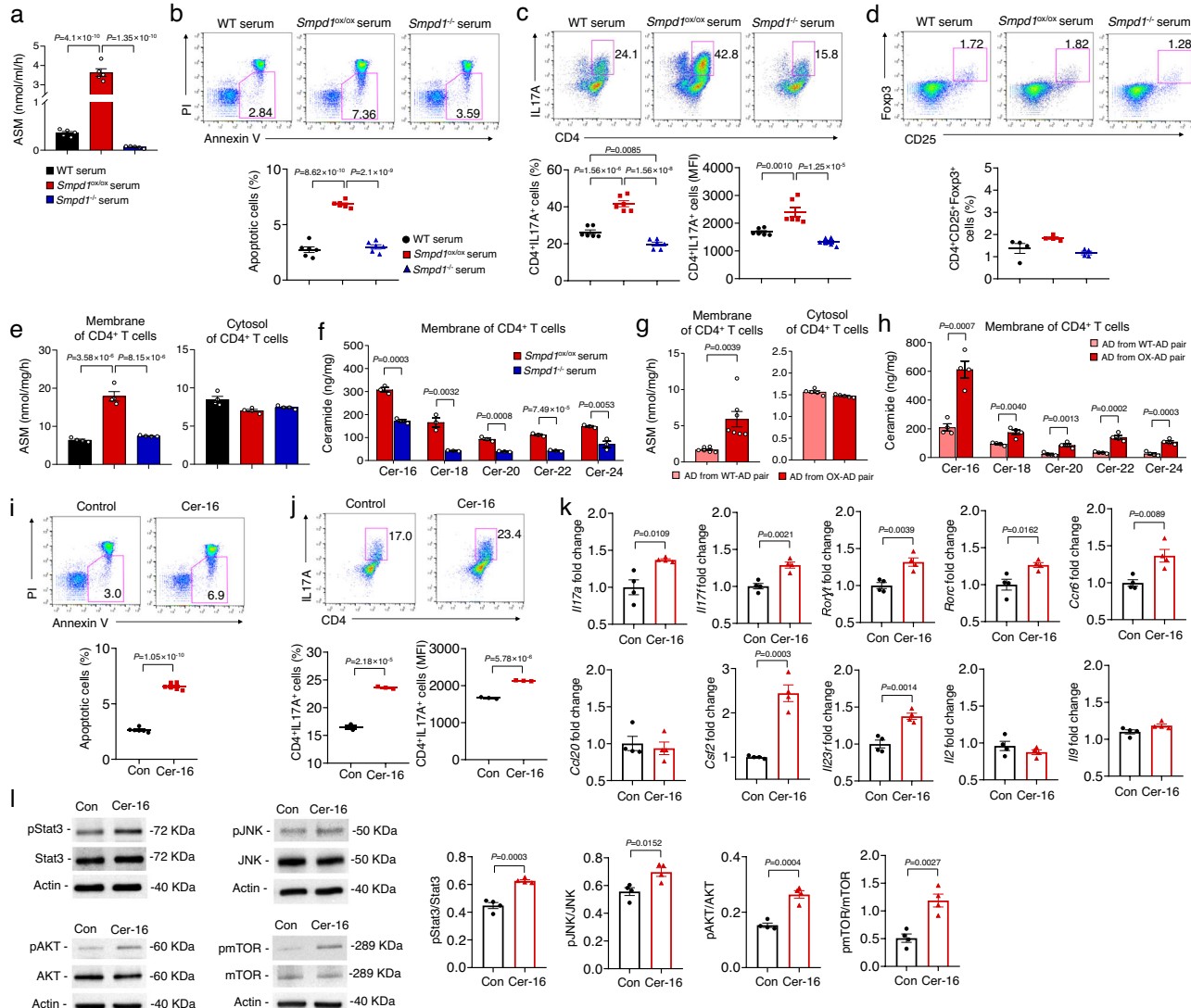

**Fig. 2 | Activated plasma ASM induces CD4+ T cell apoptosis and pathogenic Th17 differentiation by increasing ceramide in cell membranes. a** ASM activity in 3-month-old WT, *Smpd1*ox/ox (Tie2-cre; *Smpd1*ox/ox mice), or *Smpd1*−/− serum (*n* = 5/group). **b** Percentage of apoptotic cells detected with Annexin V+ in CD4+ T cell (*n* = 6/group). **c, d** Representative flow cytometry plot and graph displaying the calculated percentage and mean fluorescent intensity of Th17 cells (**c**, CD4+ IL17A+) and Treg cells (**d**, CD4+ CD25+ FoxP3+) differentiated from CD4+ T cell treated with each serum (**c** *n* = 6/group, **d** *n* = 4/group). **e** ASM activity in membrane and cytosol of CD4+ T cell treated with each serum (*n* = 4/group). **f** Ceramide concentration in membrane of CD4+ T cell treated with each serum (*n* = 3/group). **g, h** ASM activity in membrane (**g** AD from WT-AD pair, *n* = 6; AD from OX-AD pair, *n* = 7) or cytosol

(*n* = 5/group) of CD4+ T cell and ceramide levels (**h** *n* = 4/group) in membrane of CD4+ T cell. **i** Percentage of apoptotic cells detected with Annexin V+ in CD4+ T cell treated with C16 ceramide (*n* = 6/group). **j** Representative flow cytometry plot and graph displaying the calculated percentage and mean fluorescent intensity of Th17 cells differentiated from CD4+ T cell treated with C16 ceramide (*n* = 3/group). **k** mRNA levels of pathogenic Th17 cell genes in Th17 cells differentiated from CD4+ T cell by C16 ceramide treatment (*n* = 4/group). **l** Western blotting for p-Stat3, p-JNK, p-AKT, and p-mTOR in Th17 cells differentiated from CD4+ T cell by C16 ceramide treatment (*n* = 4/group). **a**–**c**, **e** One-way analysis of variance, Tukey's post hoc test. **f**–**l** Two-tailed student's *t* test. All error bars indicate s.e.m. Source data are provided as a Source Data file.

the CD4+ T cell membranes derived from these mice, but not WT mice (Fig. 2g, h).

C16-ceramide (Cer-16), which was the species most elevated in the CD4+ T cell membranes, also induced apoptosis and Th17 cell differentiation of CD4+ T cells (Fig. 2i, j), and Th17 cells differentiated by Cer-16 treatment also displayed a stronger Th17 phenotype, and most key signatures of pathogenic Th17 cells[34–36] including upregulated *Il17a*, *Il17f*, *Rorγt*, *Rorc*, and *Il23r* gene expression. Additionally, there was upregulated expression of *Csf2* and *Ccr6*, which is essential for the pathogenicity of Th17 cells[37,38] (Fig. 2k). A variety of intracellular signals including Stat3, JNK, AKT, and mTOR are critical putative elements related with Th17 cell differentiation and determine pathogenic functions of Th17 cells[39–41], and we confirmed that Cer-16 induced phosphorylation of these molecules as well (Fig. 2l).

We further investigated whether treatment with an ASM peptide (rASM) causes apoptosis and Th17 cell differentiation of CD4+ T cells similar to Cer-16. The results showed that following treatment with rASM there was an increase of ASM activity only in the CD4+ T cell membrane fraction, leading to an increase of various ceramides in the membranes as well (Supplementary Fig. 4a, b). rASM treatment also induced apoptosis and Th17 cell differentiation of CD4+ T cells by stimulating intracellular signals of Th17 cell differentiation (Supplementary Fig. 4c–e). ASM catalyzes the hydrolysis of sphingomyelin to ceramide and phosphorylcholine. To assess the involvement of phosphorylcholine in Th17 cell differentiation, we also treated CD4+ T cells with phosphorylcholine, and found no significant difference in Th17 cell differentiation (Supplementary Fig. 5).

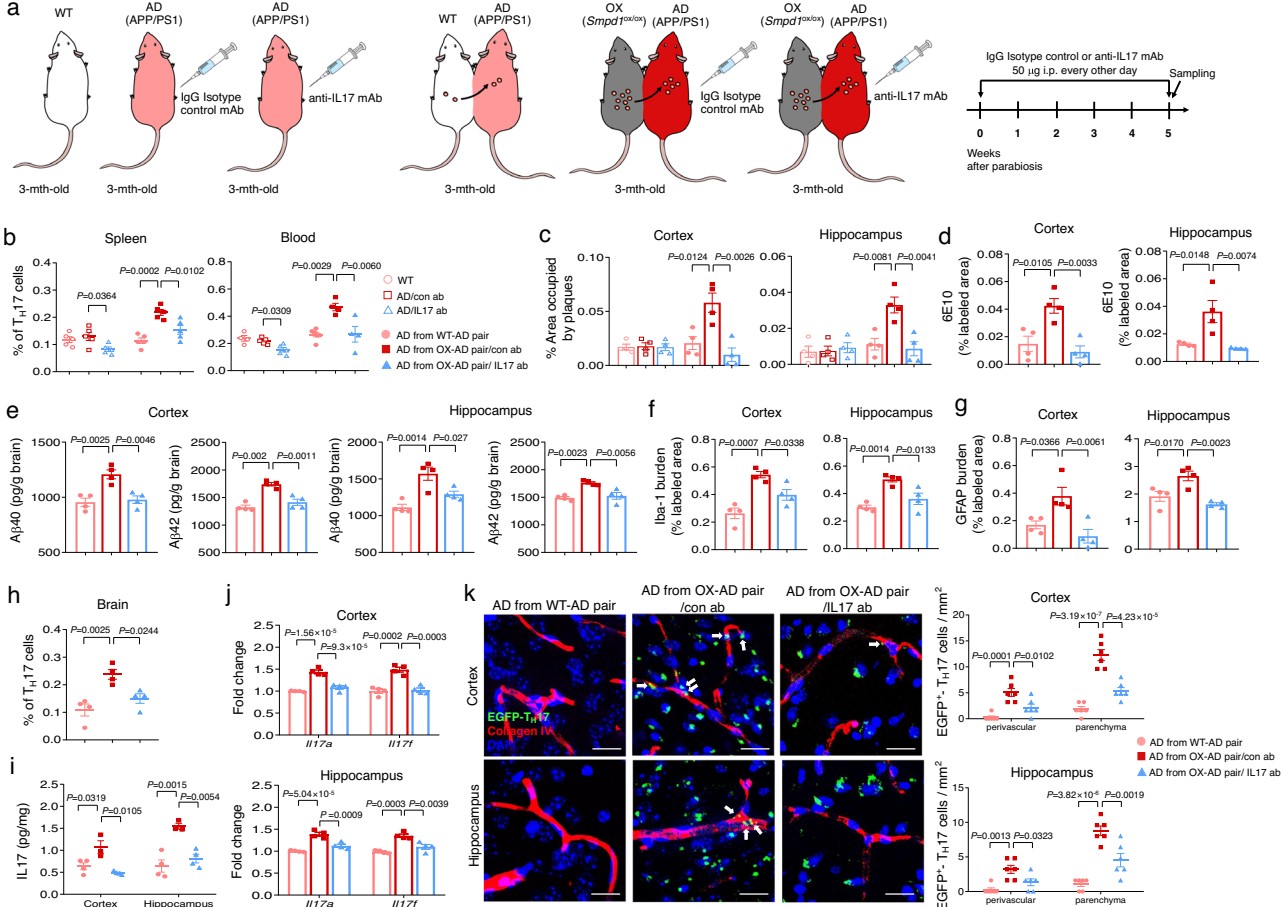

**Fig. 3 | Inhibition of blood-derived pathogenic Th17 cells prevents Aβ deposits and neuroinflammation. a** Scheme of experimental procedures. IgG isotype control or IL17 antibody was injected (50 μg/mouse, i.p. every other day) into 3-month-old of APP/PS1 mice in parabiosis with $Smpd1^{ox/ox}$ mice for 5 weeks during parabiosis or no blood exchange. **b** Percentage of Th17 cells in the spleen ($n = 5$/group) and blood (WT, AD from OX-AD pair/con ab, AD from OX-AD pair/IL17 ab, $n = 4$; AD/con ab, AD/IL17 ab, $n = 5$; AD from WT-AD pair, $n = 6$) using flow cytometry. **c** Quantification of area occupied by Aβ plaques (ThioS) ($n = 4$/group). **d** Quantification of 6E10 ($n = 4$/group). **e** Analysis of Aβ40 and Aβ42 depositions using ELISA kits ($n = 4$/group). **f**, **g** Quantification of microglia (**f**, Iba-1) and astrocyte (**g**, GFAP) ($n = 4$/group). **h** Percentage of Th17 cells in the brain using flow

cytometry ($n = 4$/group). **i**, **j** Protein levels of IL17 (**i** AD from WT-AD pair, AD from OX-AD pair/IL17 ab, $n = 4$; AD from OX-AD pair/con ab, $n = 3$) and mRNA levels (**j** $n = 4$/group) of $Il17a$ and $Il17f$. **k** Representative images and quantification of blood-derived EGFP⁺-Th17 cells (green) in the perivascular (blood vessel, collagen IV, red) and parenchyma cortex and hippocampus regions ($n = 6$/group). Scale bar: 20 μm. White arrow: perivascular EGFP⁺ Th17 cells. Extravascular cells were counted as parenchymal Th17 cells. AD, APP/PS1; OX, $Smpd1^{ox/ox}$ (Tie2-cre; $Smpd1^{ox/ox}$ mice). **b**–**k** One-way analysis of variance, Tukey's post hoc test. All error bars indicate s.e.m. All data analysis was done at 4.5-mo-old mice. Source data are provided as a Source Data file.

Collectively, these results suggested that elevated plasma ASM leads to elevated ASM activity and ceramide in CD4⁺ T cell membranes, resulting in apoptosis and pathogenic Th17 cell differentiation by stimulating downstream signals such as, Stat3, JNK, AKT, and mTOR.

### Inhibition of pathogenic Th17 cells prevents early Aβ accumulation and neuroinflammation induced by plasma ASM overexpression in young APP/PS1 mice

Based on the above results, we examined whether pathogenic Th17 cells differentiated by elevated plasma ASM promoted Aβ accumulation and neuroinflammation of young APP/PS1 mice. Three-month-old APP/PS1 mice were intraperitonially injected with IgG isotype control or IL17 antibody (50 μg per mouse, every other day) during 5 weeks of exposure to blood of ASM overexpressing Tie2-cre; $Smpd1^{ox/ox}$. APP/PS1 with no blood exchange were used as a control (Fig. 3a). IL17 antibody injection induced a decrease of Th17 cells in the spleen and blood of young APP/PS1 mice both with and without exposure to ASM overexpressing blood compared to control antibody injected mice (Fig. 3b). However, early Aβ accumulation and activation of microglia or astrocytes were prevented only in the cortex and hippocampus of

APP/PS1 mice exchanged with blood of Tie2-cre; $Smpd1^{ox/ox}$ mice (Fig. 3c–g). Although antibodies generally do not cross the BBB, IL17 antibody treatment also led to reduction of Th17 cells in the brain, as well as reduction of several pathogenic genes and proteins associated with Th17 cells (Fig. 3h–j).

To confirm whether these Th17 cells found in the brain were derived from the blood, Il17a-EGFP knockin mice were used. This mouse possess an IRES-EGFP sequence after the stop codon of the Il17a gene, so that EGFP expression is limited to IL-17A-expressing cells only. Splenic naive CD4⁺ T cells, which are EGFP⁻, were cultured in media containing rASM and Th17 differentiation cytokines to generate pathogenic EGFP⁺ Th17 cells. These cells were intravenously injected into young APP/PS1 mice exchanged with blood of Tie2-cre; $Smpd1^{ox/ox}$ mice treated with IgG isotype control or IL17 antibody. Blood-derived EGFP⁺ Th17 cells were detected both in perivascular and parenchyma regions of the cortex and hippocampus of APP/PS1 mice exposed to blood of Tie2-cre; $Smpd1^{ox/ox}$ mice treated IgG isotype control antibody compared to mice exposed to WT blood. IL17 antibody treatment led to reduction of these blood-derived cells in each region of the brain (Fig. 3k). Thus, these data revealed the critical role of blood-derived

pathogenic Th17 cells on acceleration of neuropathological features in the brain of young APP/PS1 mice exposed to overexpressed plasma ASM.

Previous studies have demonstrated that molecules-derived from pathogenic Th17 cells such as IL17 could impact pro-inflammatory responses and the phagocytic function of microglia, resulting in neuroinflammation[30,37,38]. To directly test the effects of pathogenic Th17 cells differentiated by plasma ASM on microglia, we co-cultured BV2 microglial cells with Th17 cells differentiated by ASM treatment. IL17 antibody also was used to inhibit the pathogenic functions of Th17 cells induced by the plasma ASM. The results showed upregulated pro-inflammatory cytokines and downregulated anti-inflammatory cytokine in BV2 microglial cells co-cultured with these Th17 cells, while Th17 cells simultaneously treated with ASM and IL17 antibody did not induce changes of inflammatory cytokines in BV2 microglial cells (Fig. 4a). Consistent with these results, assessments of several morphological parameters indicated that BV2 microglial cells co-cultured with the Th17 cells had a pro-inflammatory phenotype, but not the BV2 microglial cells exposed to the IL17 antibodies (Fig. 4b). We then evaluated the impact of these pathogenic Th17 cells differentiated by ASM on microglial phagocytosis using Fluor 555-labeled Aβ. Co-culture of BV2 microglial cells with the ASM treated Th17 cells revealed a deficient Fluor 555-labeled Aβ phagocytic capacity of the microglia, while BV2 microglial cells co-cultured with ASM-primed Th17 cells treated with IL17 antibody did not exhibit this abnormal function (Fig. 4c).

Similar to these in vitro results, IL17 antibody injection reduced the levels of pro-inflammatory cytokines and elevated the levels of anti-inflammatory cytokines in the cortex of young APP/PS1 mice in parabiosis with the ASM overexpressing Tie2-cre; *Smpd1*<sup>ox/ox</sup> mice (Fig. 4d). These mice also showed increased microglial recruitment surrounding Aβ and decreased parameters of a pro-inflammatory microglial phenotype (Fig. 4e, f). Additionally, we observed an increased phagocytic function of microglia following IL17 antibody injection by measuring the number of microglia containing lysosomes that co-stained with Aβ (Fig. 4g).

Several studies have revealed that cytokines released by pathogenic Th17 cell cause disruption of BBB tight junctions, resulting in increased BBB permeability and induction of efficient entry of Th17 cells into the brain parenchyma[30,42,43]. Young APP/PS1 mice exposed to the blood of the ASM overexpressing Tie2-cre; *Smpd1*<sup>ox/ox</sup> mice exhibited reduction of tight junction protein and mRNA expression (*Zo1*, *ClaudinS*, and *Occludin*), and increased BBB permeability as revealed by the accumulation of plasma-derived proteins (fibrin and thrombin) in the cortex. These BBB changes did not occur following IL17 antibody injection (Fig. 4h–j). We also observed similar protective effects of IL17 antibody injection in the hippocampus of these mice (Supplementary Fig. 6). Taken together, these data implied that pathogenic Th17 cells differentiated by overexpressed plasma ASM could efficiently migrate into the brain by causing BBB disruption, and induced inflammation and impaired phagocytic function of microglia, contributing to early Aβ accumulation and neuroinflammation in young APP/PS1 mice.

### Antibody-based inhibition of plasma ASM activity protects neuropathological features in APP/PS1 mice by blocking pathogenic Th17 cells

Next, we investigated the impact of plasma ASM inhibition on AD-related pathologies in the setting of parabiosis. Blood was shared between APP/PS1 mice and APP/PS1, WT, and *Smpd1*<sup>−/−</sup> mice at 7.5 months of age. Parabiosis was allowed to occur for 5 weeks (Supplementary Fig. 7a). Prior to analysis of the plasma ASM activity in each parabionts, we hypothesized that ASM antibodies would be produced in the blood of *Smpd1*<sup>−/−</sup> mice exposed to the blood of APP/PS1 mice because *Smpd1*<sup>−/−</sup> mice do not have ASM protein. Consistent with our hypothesis, the data showed the presence of high ASM antibody titers in the blood of the *Smpd1*<sup>−/−</sup>–APP/PS1 parabiont pair compared to those of APP/PS1–APP/PS1 and WT–APP/PS1 pairs (Supplementary Fig. 7b). In addition, the plasma ASM activity was dramatically reduced in the *Smpd1*<sup>−/−</sup>–APP/PS1 parabionts compared to the APP/PS1–APP/PS1 and WT–APP/PS1 pairs (Supplementary Fig. 7c). No changes in the levels of plasma ceramide or ASM activity in the brain tissue (cortex and hippocampus) were observed among any of the parabiont pairs (Supplementary Fig. 7d, e), indicating that the ASM antibodies present in the *Smpd1*<sup>−/−</sup>–APP/PS1 pairs efficiently reduced plasma ASM activity but did not reach the brain tissue. There also was reduced ASM activity and ceramide levels in the CD4<sup>+</sup> T cell membranes of the *Smpd1*<sup>−/−</sup>–APP/PS1 pairs (Supplementary Fig. 7f, g), leading to a decrease of pathogenic Th17 cells and increase of Treg cells in the spleen and blood (Supplementary Fig. 7h, i).

Pathogenic Th17 cells can induce inflammatory responses or chemotaxis of other immune cells, such as monocytes, macrophages, and neutrophils[30,40,41]. We therefore analyzed changes of other immune cell populations including monocyte, macrophages, and neutrophils in these mice. Although neutrophils were increased in the blood of *Smpd1*<sup>−/−</sup> blood-exposed APP/PS1 mice, pro-inflammatory Ly6c<sup>hi</sup> monocytes and macrophages were reduced and anti-inflammatory Ly6c<sup>low</sup> monocytes was increased (Supplementary Fig. 7j). Similar to the spleen and blood results, a decrease of pathogenic Th17 cells also was observed in the perivascular and parenchymal brain regions of these mice through Rorγ staining and effusion of FITC-labeled albumin, while Treg cells increased (Supplementary Fig. 7k–m). Monocytes, M1-like macrophages, and neutrophils also were diminished, and M2-liked macrophages were elevated in the brain (Supplementary Fig. 7n). Together, these data indicated that inhibition of plasma ASM activity by ASM antibodies present in the *Smpd1*<sup>−/−</sup>–APP/PS1 pairs led to a decrease of pathogenic Th17 cells in the spleen, blood, and brain, and this might contribute to reduction of pro-inflammatory immune cells and increase of anti-inflammatory immune cells in these mice.

In accordance with a decrease of pathogenic Th17 in the perivascular brain region of these mice, the damage normally found in the BBB (disruption of BBB tight junction, permeability, and vessels structure) also was reduced (Supplementary Fig. 8). As a result, microglia and astrocyte activation were reduced in the cortex and hippocampus (Supplementary Fig. 9a, b), and the pro-inflammatory state and deficient phagocytic function of the microglia were restored (Supplementary Fig. 9c–f). Furthermore, these protective effects led to prevention of Aβ accumulation, cerebral amyloid angiopathy, and even synapse loss in APP/PS1 mice (Supplementary Fig. 10). Therefore, these "proof-of-concept" findings suggested that antibody-based inhibition of plasma ASM activity could prevent BBB disruption, neuroinflammation, Aβ accumulation, and synapse loss by decreasing pathogenic Th17 cells in the blood and brain of APP/PS1 mice.

### Plasma ASM-targeting immunotherapy prevents neuropathological changes in APP/PS1 mice

To further verify the impact of antibody-based inhibition of plasma ASM activity on AD pathologies, 6-month-old WT or APP/PS1 mice were immunized with a recombinant ASM peptide in adjuvant or adjuvant containing a peptide vehicle (PBS) 4 times until 9 months of age. Blood from each mouse were extracted before immunization or after the 2nd, 3rd, and 4th immunizations, respectively (Fig. 5a). As expected, we observed the presence of high ASM antibody titers and inhibition of ASM activity in the plasma of ASM-immunized WT and ASPP/PS1 mice at each post-immunization time point (Fig. 5b). However, as observed in the parabionts, plasma ceramide levels, as well as ASM activity in the cortex and hippocampus of these mice, were not altered (Fig. 5c, d). In addition, we found that the ASM activity and ceramide levels were significantly decreased in the CD4<sup>+</sup> T cell

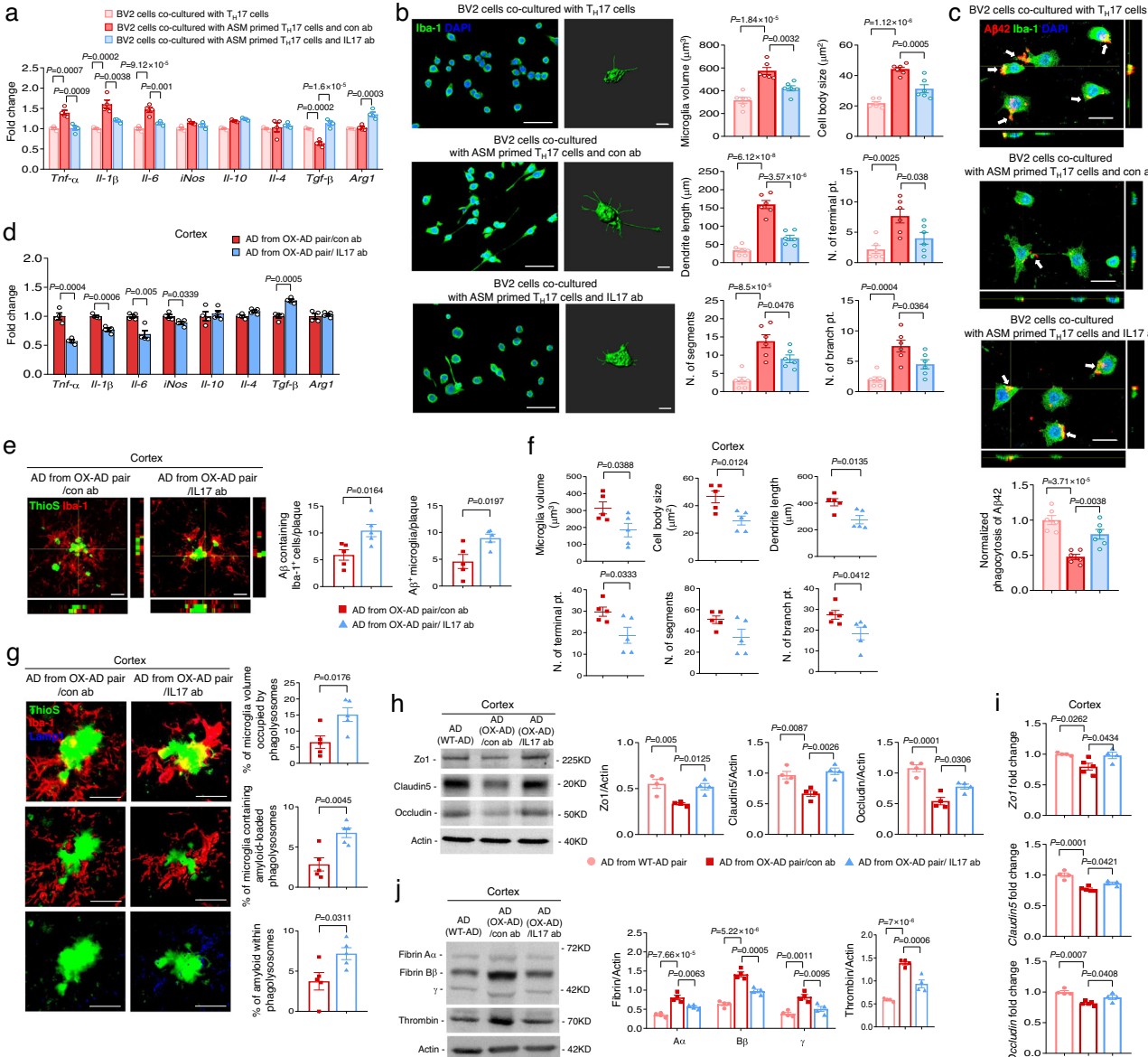

**Fig. 4 | Inhibition of blood-derived pathogenic Th17 cells protects microglia phagocytic function defects and reduces BBB disruption in the cortex of young APP/PS1 mice in parabiosis with *Smpd1*<sup>ox/ox</sup> mice. a** mRNA levels of inflammatory markers in BV2 microglial cells (*n* = 4/group). **b** Left, immunofluorescence images (Iba-1, Scale bars, 50 μm) and imaris-based three-dimensional images (Scale bars, 10 μm) of BV2 microglial cells. Right, imaris-based automated quantification of microglial morphology (*n* = 6/group). **c** Top, immunofluorescence images of microglia with Fluor 555-labeled Aβ. Scale bars = 50 μm. Bottom, quantification of the Fluor 555-labeled Aβ uptake (*n* = 6/group). **d** mRNA levels of inflammatory markers in the cortex (*n* = 4/group). **e** Immunostaining images and quantification of Aβ (ThioS, green) positive cells and microglia (Iba1, red) (*n* = 5/group). Scale bars = 10 μm. **f** Imaris-based automated quantification of microglial morphology (*n* = 5/group). **g** Left, immunofluorescence images of ThioS (Aβ plaques, green)

encapsulated within Lamp1+ structures (phagolysosomes, blue) in microglia (Iba1, red) present in the cortex. Scale bars, 20 μm; 3D reconstruction from confocal image stacks scale bars, 10 μm. Right, quantification of microglia volume occupied by Lamp1+ phagolysosomes, percent of microglia containing Aβ-loaded phagolysosomes and Aβ encapsulated in phagolysosomes (*n* = 5/group). **h** Western blot analysis and quantification of tight junction proteins (Zo1, Claudin5, Occludin) in the cortex (*n* = 4/group). **i** mRNA levels of tight junction (AD from WT-AD pair, AD from OX-AD pair/IL17 ab, *n* = 4; AD from OX-AD pair/con ab, *n* = 5). **j** Western blot analysis and quantification of fibrin and thrombin (*n* = 4/group). **a–c, h–j** One-way analysis of variance, Tukey's post hoc test. **d–g** Two-tailed student's *t* test. All error bars indicate s.e.m. All data analysis was done at 4.5-mo-old mice. Source data are provided as a Source Data file.

membrane fractions of ASM-immunized APP/PS1 mice (Fig. 5e), resulting in a decrease of pathogenic Th17 cells and increase of Treg cells in the spleen and blood (Fig. 5f, g). Additionally, immunization with the ASM peptide normalized the pro-inflammatory monocyte and macrophages in the blood of APP/PS1 mice, but not neutrophils (Fig. 5h). ASM-immunized APP/PS1 mice also showed reduction of pathogenic Th17 cells, monocytes, and even neutrophils in the brain, and increase of Treg cells and M2-like macrophages compared to vehicle-treated APP/PS1 mice (Fig. 5i–l). These data indicated that

inhibition of plasma ASM activity following immunization of the APP/PS1 mice with an ASM peptide reduced pathogenic Th17 cells in the spleen, blood, and brain, and this was accompanied by normalization of other immune cell populations altered in these mice.

Immunization with the ASM peptide exhibited a protective effect on tight junction loss and BBB permeability in APP/PS1 mice (Supplementary Fig. 11a–d). In vivo time-lapse multiphoton imaging also showed obvious preventive effects on BBB permeability in the ASM-immunized APP/PS1 mice compared to vehicle-treated mice (Fig. 5m),

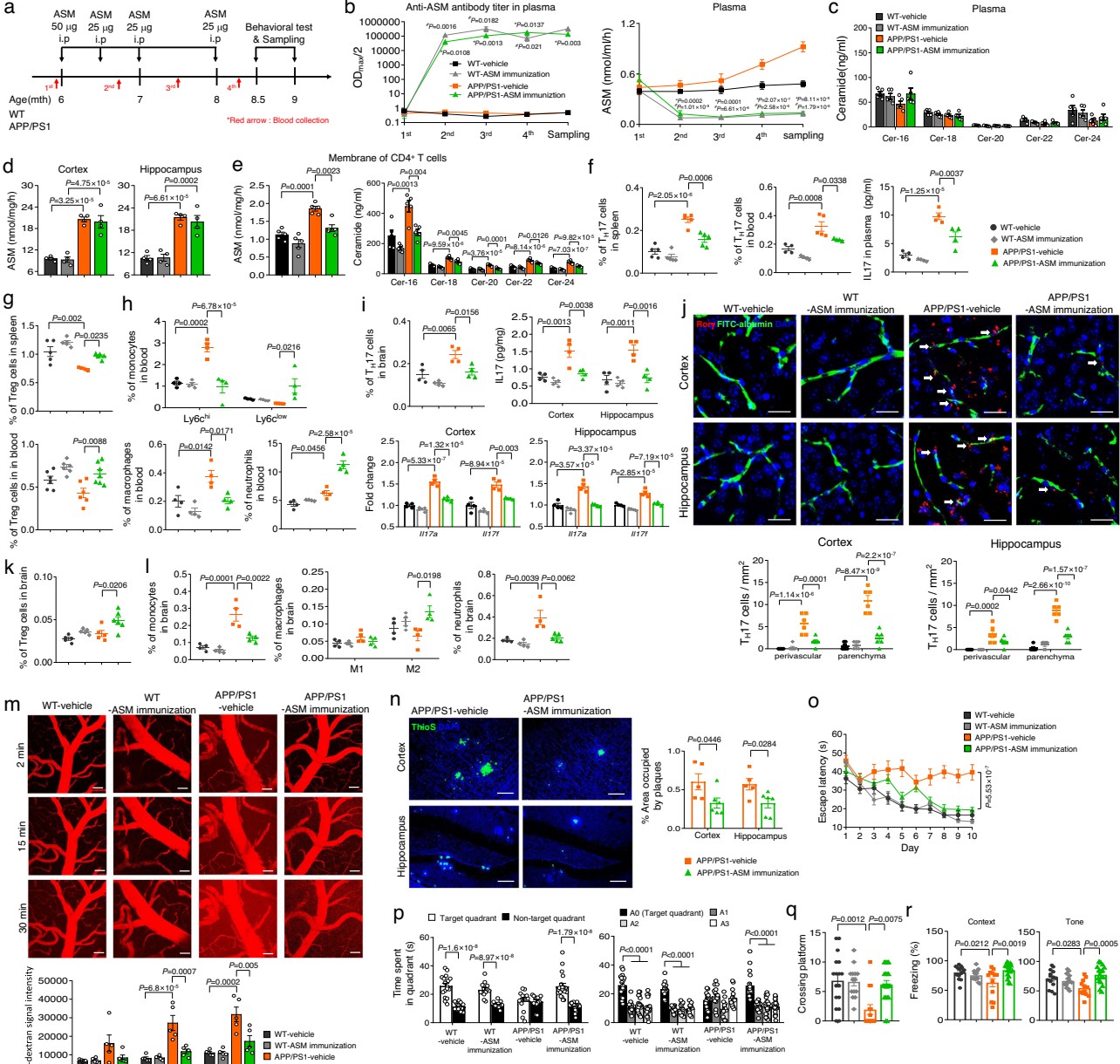

**Fig. 5 | Immunization with an ASM peptide prevents BBB disruption and memory impairment in APP/PS1 mice. a** Scheme of experimental procedures. **b** Anti-ASM antibody titers (*n* = 8/group) and ASM activity (*n* = 8/group). #*P*: WT-vehicle vs. WT-ASM, *P: APP/PS1-vehicle vs APP/PS1-ASM. **c** Ceramide concentration. **d** ASM activity. **e** ASM activity and ceramide levels in the CD4+ T cell membranes (**c**, **e** *n* = 5/group; **d** *n* = 4/group). **f, g** Percentage of Th17 and Treg cells in the spleen (*n* = 5/group) and blood (**f** WT-vehicle, WT-ASM, APP/PS1-ASM, *n* = 4; APP/PS1-vehicle, *n* = 5, **g** WT-vehicle, WT-ASM, APP/PS1-vehicle, *n* = 6; APP/PS1-ASM, *n* = 7) and IL17 protein levels in plasma (*n* = 4/group). **h, i** Percentage of pro (Ly6c[hi])- or anti (Ly6c[low])-monocytes, macrophages, and neutrophils in the blood, Th17 cells in the brain, IL17 protein levels and *Il17a* and *Il17f* mRNA levels (**h, i** *n* = 4/group). **j** Immunofluorescence images and quantification of Th17 cells (RORγ, red; blood vessel, FITC-labeled albumin, green) (*n* = 6/group). Scale bars, 20 μm. White arrow: perivascular Th17 cells. **k, l** Percentage of Treg cells (**k** WT-vehicle, WT-ASM, APP/

PS1-vehicle, *n* = 5; APP/PS1-ASM, *n* = 6), monocytes (**l** *n* = 4/group), M1 or M2 macrophages (**l** *n* = 4/group), and neutrophils (l WT-vehicle, WT-ASM, APP/PS1-vehicle, *n* = 4; APP/PS1-ASM, *n* = 5) in the brain. **m** Multiphoton images and quantification of tetramethylrhodamine (TMR) dextran (red) leakage from cortical vessels (*n* = 5/group). Scale bars, 50 μm. **n** Immunofluorescence images and quantification of ThioS (Aβ plaques). Scale bars, 50 μm. (APP/PS1-vehicle, *n* = 5; APP/PS1-ASM, *n* = 6). **o** The results of the Morris water maze test. **p, q** Time spent in target platform (**p**) and the number of times entered the small target zone (**q**) at probe trial day 11 (**o–q** WT-vehicle, *n* = 17; WT-ASM, APP/PS1-vehicle, *n* = 14; APP/PS1-ASM, *n* = 20). **r** The results of fear-conditioning test (WT-vehicle, WT-ASM, *n* = 14; APP/PS1-vehicle, *n* = 12; APP/PS1-ASM, *n* = 17). **d–m, o–r** One-way analysis of variance, Tukey's post hoc test. **b, n, p** Two-tailed student's t test. All error bars indicate s.e.m. Source data are provided as a Source Data file.

and there was less damage of microvascular structures in the cortex and hippocampus as well (Supplementary Fig. 11e, f). The decrease of pathogenic Th17 cells in the brain of these immunized APP/PS1 mice resulted in prevention of microglia and astrocyte activation, changes of inflammatory cytokines, morphological alteration of microglia, and

deficient microglia phagocytic function (Supplementary Fig. 12a–e). Consequently, Aβ accumulation was lower in the cortex and hippocampus (Fig. 5n and Supplementary Fig. 12f–h), leading to improvements of synapse loss in ASM-immunized APP/PS1 mice and learning and memory in these male mice (Supplementary Fig. 12i, j and

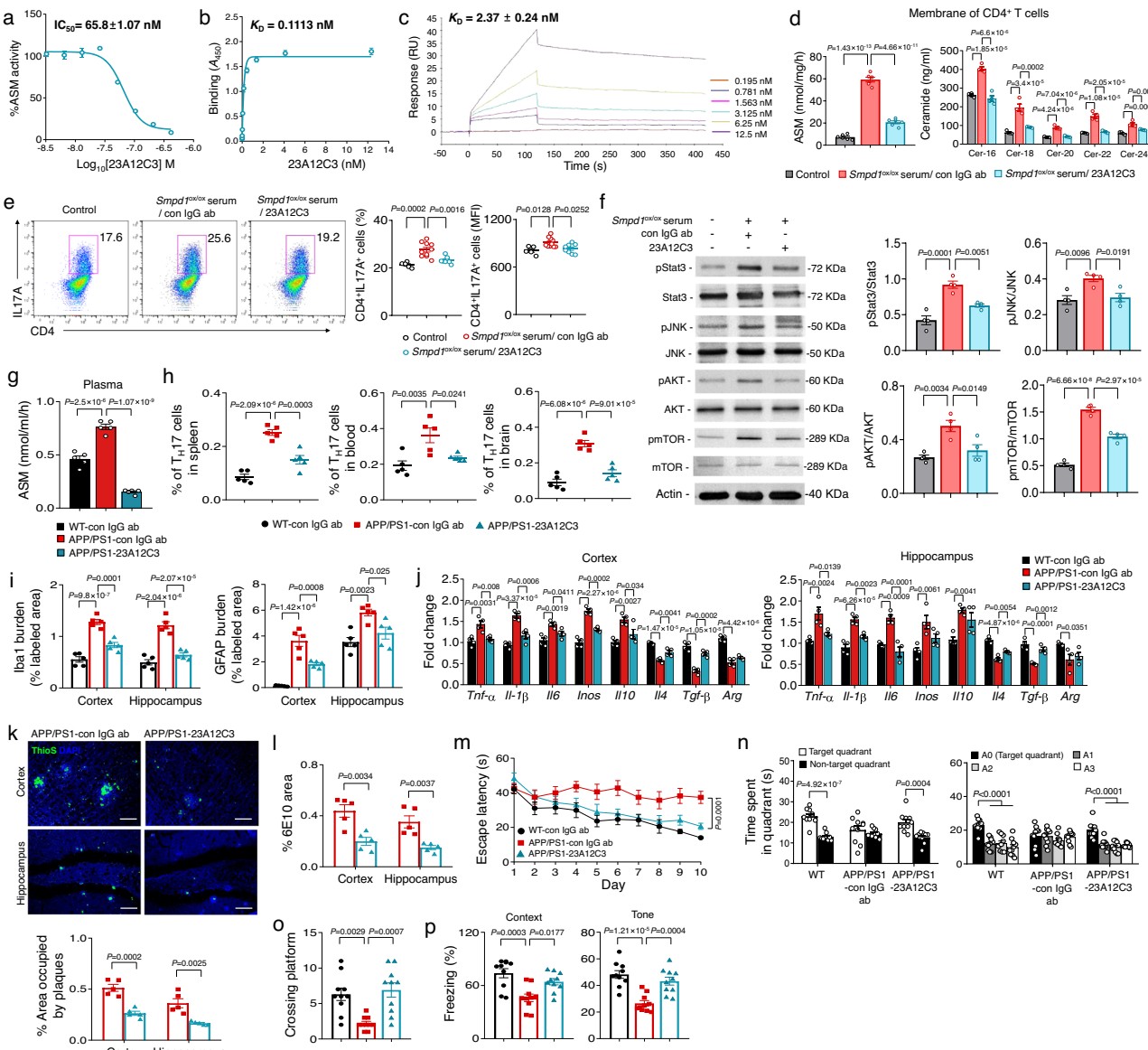

**Fig. 6 | Monoclonal ASM antibody (23A12C3) inhibits pathogenic Th17 cell differentiation and protects neuropathological features in APP/PS1 mice.**
**a** Effect of various concentrations of 23A12C3 on the direct inhibition of ASM activity ($n = 3$ independent experiments). **b** Binding affinity ($K_D$) of 23A12C3 with recombinant human ASM by ELISA ($n = 6$ independent experiments).
**c** Representative binding sensorgrams depicting the interaction of 23A12C3 with recombinant human ASM assessed by SPR detection. **d** ASM activity ($n = 6$/group) and ceramide concentration ($n = 4$/group) in membrane of CD4[+] T cells treated with *Smpd1*[ox/ox] serum (from 3-month-old Tie2-cre; *Smpd1*[ox/ox] mice) and IgG isotype control or 23A12C3. **e** Representative flow cytometry plot and graph displaying the calculated percentage and mean fluorescent intensity of Th17 cells (CD4[+] IL17A[+]) differentiated from CD4[+] T cell in each group (Control, $n = 5$; *Smpd1*[ox/ox] serum/con IgG ab, $n = 10$; *Smpd1*[ox/ox] serum/23A12C3, $n = 8$). **f** Western blotting for p-Stat3, p-JNK, p-AKT, p-mTOR in Th17 cells differentiated from CD4[+] T cell in each group

($n = 4$/group). **g** ASM activity in plasma ($n = 5$/group). **h** Percentage of Th17 cells in the spleen, blood, and brain by flow cytometry ($n = 5$/group). **i** Quantification of microglia (Iba-1) and astrocyte (GFAP) ($n = 5$/group). **j** mRNA levels of inflammatory markers in the cortex and hippocampus ($n = 4$/group). **k** Representative immuno-fluorescence images and quantification of ThioS (Aβ plaques) in cortex and hippocampus. Scale bars, 50 μm. ($n = 5$/group). **l** Quantification of 6E10 ($n = 5$/group). **m** The results of Morris water maze test ($n = 10$/group). **n, o** At probe trial day 11, time spent in target platform (**n**) and the number of times each animal entered the small target zone (**o**) were analyzed ($n = 10$/group). **p** The results of fear-conditioning test ($n = 10$/group). **d–j, m–p** One-way analysis of variance, Tukey's post hoc test. **k, l, n** Two-tailed student's t test. All error bars indicate s.e.m. All in vivo data analysis was done at 9-mo-old mice. Source data are provided as a Source Data file.

Fig. 5o–r). Overall, these data revealed the significant prophylactic effects of immunization of APP/PS1 mice with an ASM peptide on various pathological features, including changes of immune cell populations, BBB damage, neuroinflammation, Aβ accumulation, synapse loss, and learning and memory impairment.

Finally, to further examine the prophylactic effects of immunization on the neuropathological changes in AD, we generated mouse monoclonal antibodies to ASM. The 23A12C3 antibody was selected by

in vitro screening with an ASM activity assay, and it was evaluated for its ability to block ceramide production by the recombinant ASM protein. 23A12C3 showed high ASM inhibitory potency and selective binding to ASM (Fig. 6a, b). The result of biosensor experiments using surface plasmon resonance (SPR) detection also revealed high binding and dose dependent affinity to ASM (Fig. 6c).

Prior to testing the effects of this antibody in vivo, we first confirmed the inhibitory effects of 23A12C3 on ASM activity and ceramide

in the CD4[+] T cell membranes. The serum from Tie2-cre; *Smpd1*[ox/ox] mice led to increased ASM activity and ceramide levels in the cell membranes, but these elevations were decreased by 23A12C3 treatment (Fig. 6d). As a result, differentiation of pathogenic Th17 cells and related downstream signals were reduced (Fig. 6e, f). Next, we intraperitoneally administered 23A12C3 (50 mg kg⁻¹) into 7-month-old of APP/PS1 mice twice a week for 8 weeks. 23A12C3 efficiently decreased plasma ASM activity (Fig. 6g), leading to reduction of pathogenic Th17 cells in the spleen, blood, and brain, and further showed preventive effects on microglia and astrocytes activation, increase of pro-inflammation, Aβ accumulation in APP/PS1 mice, and on memory impairment in these male mice (Fig. 6h–p). Collectively, these results demonstrated that active or passive immunization targeting plasma ASM suppressed the pathogenic differentiation of Th17 cells in the blood, and protected against several neuropathological features in AD mice.

## Discussion

Several studies have demonstrated a correlation between elevated ASM activity and various neurodegenerative disease pathologies, including those found in AD. Although some of these correlations have been attributed to the production of sphingomyelin-derived ceramide from ASM, the specific role of this enzyme in many of these pathologies has not been fully elucidated. In this study, we reveal several roles for plasma ASM in AD. In agreement with previous reports, we first documented that plasma ASM activity in AD patients and APP/PS mice was elevated and showed a positive correlation with disease progression or age. Parabiosis experiments using young AD mice and mice overexpressing plasma ASM indicated that the plasma ASM contributed to early Aβ accumulation, pro-inflammation, and deficient microglia phagocytic function in the brain, despite the fact that there was no increase of ASM activity in the brain tissue itself. These results suggested a specific role of plasma ASM in the acceleration of AD brain pathology.

As described in several previous studies[28–30] showing changes in the peripheral immune system of AD patients or mice that contributed to neuroinflammation in the brain, we observed reduction of CD4[+] T and Treg cells, and an increase of Th17 cells, in the spleen and blood of young AD mice exposed to high plasma ASM. These data indicated that plasma ASM impacted survival of CD4[+] T cells and differentiation into Th17 cells in the blood. Secretory forms of ASM, including plasma ASM, are known to primarily localize to the extracellular space or outer surface of the cell membrane to catalyze hydrolysis of membrane-bound sphingomyelin to ceramide[19]. Since plasma ceramide levels were not changed in the blood of young AD mice exposed to high plasma ASM, we investigated other mechanisms by which increased plasma ASM mediates the apoptosis and Th17 cell differentiation of CD4[+] T cells. We found an elevation of ASM activity and ceramide levels in CD4[+] T cell membranes, but not cytosol, exposed to activated plasma ASM in vitro and in vivo. These results showed that ASM increased in the plasma was recruited into the CD4[+] T cell membranes, leading to CD4[+] T cell apoptosis and pathogenic Th17 cell differentiation though ceramide production in the cell membrane. This result was also supported by previous studies demonstrating that the ASM/ceramide pathway has a role in T cell apoptosis, activation, and differentiation[39,44]. Although activation of the ASM/ceramide pathway did not directly affect Treg cell differentiation in these in vitro experiments, we observed a decrease of Treg cells in the spleen and blood of young AD mice exposed to the elevated plasma ASM. The differentiation fate of CD4[+] T cells into T helper or Treg cells is mainly determined by downstream signals, such as activation of Stat3 signals that promote Th17 differentiation and inhibit the generation of Treg cells[39–41]. In this regard, we confirmed that activation of the ASM/ceramide pathway elevated the phosphorylation of Stat3, as well as JNK, AKT, and mTOR, which are other key signals related to Th17 cell differentiation. Therefore, the decrease of Treg cells in the spleen and

blood might be a concomitant consequence of activated downstream signals related to Th17 cell differentiation, induced by elevated plasma ASM in the young AD mice.

We further found that the elevated plasma ASM induced expression of *Il17a*, *Il17f*, *Roryt*, *Rorc*, *Csf2* and *Ccr6*, factors that likely contribute to the increase of pathogenic Th17 cells in the blood of young AD mice. These cytokines, especially IL17A, IL17F, and CSF2, are also capable of promoting BBB disruption by reducing tight junction expression on endothelial cells, contributing to an increase of BBB permeability and favoring the entrance into the brain of parenchymal Th17 cells. Moreover, these cytokines are able to enter the brain and negatively affect brain function. Th17 cells that directly infiltrate into the brain will promote expression of genes encoding pro-inflammatory microglia or chemotaxis of other immune cells[30,45], leading to brain pathology. Previous studies have shown the critical roles of IL17 and related cytokines in promoting AD neuroinflammation and neurodegeneration, and neutralization of IL17 ameliorates cognitive impairment and Aβ-induced neuroinflammation in adult mice[46,47]. In this study, we also observed that the administration of antibodies against IL17 prevented many of the pathologies induced in young AD mice that were overexpressing ASM, including BBB disruption, the formation of the pro-inflammation morphology, and the deficient phagocytic function of microglia. Moreover, the restoration of microglia phagocytic function by the injected anti-IL17 antibodies inhibited early Aβ accumulation in these mice, indicating that pathogenic Th17 cells differentiated by the activated plasma ASM/ceramide pathway play a critical role in acceleration of AD brain pathology in young AD mice. Recently, an increase of Th17 cells has been observed in the blood of mild cognitively impaired AD patients compared to control subjects, and there was a positive association found between the level of Th17 cells and amyloidopathy as well[48]. Other studies in both humans and animal models have shown the relevance of Th17 cells in early AD[49], and are in agreement with our results showing an increase of plasma ASM activity in early AD patients resulted in infiltration of Th17 cells into the brain and an accelerated brain pathology.

There are increasing efforts underway to develop an antibody-based immunotherapeutic approach for various neurodegenerative diseases. The advantages of this approach include high target specificity, reduced off-target side effects, and a known pharmacokinetic profile. Evaluation of immunotherapy for AD has shown that targeting Aβ, tau, and BACE1 with antibodies can improve pathology in both mouse models or human brain[50–54]. However, limited amounts of circulating antibodies cross the BBB and enter the brain, and there is some uncertainty as to whether the positive therapeutic effects are due to antibodies that have reached the brain or are still present in the blood. Therefore, these studies remain somewhat controversial. In principle, based on our studies an immunotherapy targeting plasma ASM could overcome the limitation of crossing the BBB since its primary site of action would remain in the blood, not the brain tissue. In support of this concept, the results from parabiosis experiments using *Smpd1*[−/−] mice and active or passive immunotherapy in AD mice showed efficient inhibition of plasma ASM activity by ASM antibodies. Furthermore, we show marked prophylactic effects of the ASM antibody on neuropathological changes in AD despite no decrease of ASM activity in the brain. These results highlight the potential of this immunotherapeutic approach targeting plasma ASM for AD.

Notably, the inhibition of plasma ASM activity by the anti-ASM antibody also suppressed the pathogenic differentiation of Th17 cells in the blood, and contributed to normalization of other immune cell populations that affect brain neuroinflammation. This impact on pathogenic Th17 cells and other immune cell alterations is likely to contribute to the prevention of BBB disruption, neuroinflammation, Aβ accumulation, synapse loss, and even learning and memory impairment in the AD mice we observed (Supplementary Fig. 13). Thus, our findings suggested that plasma ASM-targeted immunotherapy

could be a next generation immunotherapy for AD and, importantly, unlike other immunotherapies, does not require BBB penetration. Although we focused on the prophylactic effects of immunotherapy targeting plasma ASM in this study, we think that the ASM antibody also may have therapeutic effects in ameliorating neuropathological features in advanced AD mice as well. For this, we will be conducting further studies in the future. Elevated activity of plasma ASM is found in a variety of diseases including heart disease, diabetes mellitus, and inflammatory diseases such as sepsis and systemic inflammatory response syndrome[19]. If the pathological roles of plasma ASM is revealed in these diseases, immunotherapy targeting plasma ASM could be useful in these diseases as well.

# Methods

## Mice

The following mouse lines were used: C57BL/6 wild type (WT) mice (The Jackson Laboratory), Tie2-Cre mice (stock number 008863, The Jackson Laboratory), loxP-flanked Tg-$Smpd^{stop}$ mice[22] ($Smpd1^{ox/ox}$ mice, C57BL/6 background), $Smpd1^{-/-}$ mice[33] (C57BL/6 background), and Il17a-EGFP knockin mice (stock number 018472, The Jackson Laboratory). To obtain endothelial cell-specific ASM overexpressing mice, Tg-$Smpd^{stop}$ mice were crossed with Tie2-Cre mice. Transgenic mouse lines overexpressing the hAPP695swe (APP) and presenilin-1M146V (PS1) mutations were originated from GlaxoSmithKline (Harlow, UK)[25] and maintained as described previously[18,21]. We used littermate mice that were sex- and age-matched between experimental groups. Both male and female mice were used for all experiments except behavioral studies using male mice.

The block randomization method was used to allocate the animals to experimental groups. To eliminate the bias, all investigators were blinded to the experimental groups and analysis such as data collection and data analysis. Mice were housed at a 12 h day/12 h night cycle, 21–22 °C and 50–60% humidity with free access to water and food pellets. All protocols were approved by the Kyungpook National University Institutional Animal Care and Use Committee (IACUC).

## Parabiosis and IL17 antibody treatment

Parabiosis surgery followed previously described procedures[7,12]. To investigate the effects of overexpressed plasma ASM on AD pathologies, 3-month-old of APP/PS1 and age- and weight-matched WT or $Smpd1^{ox/ox}$ (Tie2-cre; $Smpd1^{ox/ox}$) mice were selected for parabiosis surgery. IgG Isotype control (50 μg/mouse, R&D system, MAB0006) or IL17 antibody (50 μg/mouse, R&D system, MAB421) was injected intraperitoneally into APP/PS1 mice in parabiosis with WT or $Smpd1^{ox/ox}$ mice every other day for 5 weeks during parabiosis. To confirm the effects of antibody-based plasma ASM inhibition on AD pathologies, 7.5-month-old of APP/PS1 and age- and weight-matched APP/PS1, WT, or $Smpd1^{-/-}$ mice were selected for parabiosis surgery. Mirror-image incisions at the left and right flanks of age- and weight-matched mice were made through the skin, and shorter incisions were made through the abdominal wall. The peritoneal openings of the adjacent parabionts were sutured together. Elbow and knee joints from each parabiont were sutured together, and the skin of each mouse was stapled (9-mm Autoclip, Clay Adams) to the skin of the adjacent parabiont. Each mouse was injected subcutaneously with Baytril antibiotic and monitored during recovery. Five weeks after surgery, mice were sacrificed for analysis.

## Mouse plasma or serum collection

Mouse blood was collected into sodium heparin-coated tubes via intracardial bleed at the time of death. Plasma was generated by centrifugation (15,493 × g, 4 °C, 5 min) of freshly collected blood and aliquots were stored at −80 °C until use. To collect serum, mouse blood was collected in a e-tube and the blood was allowed to clot by leaving it undisturbed at room temperature for 30 min. Serum was collected by centrifugation (15,493 × g, 4 °C, 5 min) and aliquots were stored at −80 °C until use.

## Human plasma collection

Human plasma samples were obtained from both men and women with AD and age-matched non-AD controls from Hanyang University Hospital (Supplementary Table 1). Sex of participants was determined based on self-report. Informed consent was obtained from all subjects according to the ethics committee guidelines at the Hanyang University Hospital (IRB no. HYUH 2016-12-029-003).

## Preparation of brain microvessels

Brain microvessels were prepared as previously described[17]. To prepare the brain samples, mice were anesthetized and blood was collected at the time of death into EDTA-coated tubes via intracardial bleed. After blood collection, mice were transcardially perfused with PBS and brains were further removed from the skull. The meninges and choroid plexuses were removed, and the brainstem and cerebellum were dissected away from the cerebrum. Following dissections, tissues were rinsed, and diced into small pieces (approx. 1 mm). Each tissue was then homogenized in Dounce tissue grinder. Resulting homogenates were centrifuged at 1000 × g for 5 min. Pelleted material was then resuspended in 18% (w/v) dextran solution and centrifuged at 4400 × g for 15 min. The pellet (brain vessel) was carefully separated from the supernatant (vessel-depleted brain). The pellet, containing the crude vascular fraction, was resuspended in Hank's Balanced Salt Solution (HBSS). Vascular suspensions were next passed through a 100 μm nylon mesh filter to eliminate the larger, macrovascular components. The resulting filtrates were passed through a 40 μm cell strainer. The unfiltered microvessels were harvested by washing into a low binding tube and resuspended in HBSS. The brain microvessels were then resuspended in ice-cold lysis buffer, sonicated, centrifuged at 20,000 × g for 20 min, and supernatant was used for ASM activity and ceramide analysis.

## ASM activity assays

We performed the enzymatic activity measurements as previously described[17,21] using a UPLC system (Waters). Briefly, the brain was lysed in homogenization buffer containing 50 mM HEPES (Sigma-Aldrich, H3375), 150 mM NaCl (Sigma-Aldrich, S3014), 0.2% Igepal CA-630 (Sigma-Aldrich, I8896), and protease inhibitor (Calbiochem, 539131). Three microliters of the samples (plasma, serum, or brain) were mixed with 3 μl of 200 μM Bodipy-C12-sphingomyelin (Invitrogen, D7711) diluted in 0.2 M of sodium acetate buffer, pH 5.0, 0.2 mM ZnCl2, and 0.2% Igepal CA-630 and incubated at 37 °C for 1 h. The hydrolysis reactions were stopped by adding 114 μl of ethanol, and centrifuged at 15,493 × g for 5 min. Thirty microliters of the supernatant was then transferred to a sampling glass vial and 5 μl was applied onto a UPLC system for analysis. Quantification was achieved by comparison to Bodipy-C12-ceramide using the Waters Millennium software.

## LC-MS/MS for ceramide quantification

We simultaneously analyzed Cer-16, Cer-18, Cer-20, Cer-22, and Cer-24 using a Agilent 6470 triple quadrupole liquid chromatography-mass spectrometry (LC–MS/MS) system (Agilent, Wilmington, DE, USA) by the modified method of previous study[55]. Briefly, standard calibration curves for five ceramides were prepared using the mixtures of Cer-16 (Avanti, 860516), Cer-18 (Avanti, 860518), Cer-20, (Avanti, 860520), Cer-22 (Avanti, 86051), and Cer-24 (Avanti, 860524) stock solution in the range of 0.5–1000 ng ml$^{-1}$ of ceramides in HBSS. Standard curves for Cer-16, Cer-18, Cer-20, Cer-22, and Cer-24 showed good linearity ($r^2 > 0.996$ for all ceramides) and the coefficient of variance for the interday precision and accuracy was below 15 %. For the ceramide analysis, aliquots (50 μl) of plasma and protein samples extracted from CD4$^+$ T cell membrane were added to 200 μl of internal standard

solution (IS; berberine 0.1 ng ml$^{-1}$ in methanol) and vortexed for 5 min. After centrifugation of the mixture at $16,000 \times g$ for 5 min, aliquots (5 µl) of the supernatant were injected into a LC–MS/MS system. Cer-16, Cer-18, Cer-20, Cer-22, and Cer-24 and berberine (IS) was separated on a Synergi Polar RP column (2.0 × 150 mm, 4 µm particle size; Phenomenex, Torrence, CA, USA) using a isocratic elution of distilled water containing 0.1% formic acid: methanol containing 0.1% formic acid = 10:90 (v/v) at a flow rate of 0.25 ml min$^{-1}$. Quantification of the analyte peaks was carried out at m/z 538.5 → 264.3 for Cer-16 (retention time ($T_R$) 4.6 min), m/z 566.5 → 264.3 for Cer-18 ($T_R$ 5.2 min), m/z 594.6 → 264.3 for Cer-20 ($T_R$ 6.3 min), m/z 622.6 → 264.3 for Cer-22 ($T_R$ 7.5 min), m/z 650.7 → 264.3 for Cer-24 ($T_R$ 9.2 min), and m/z 336.1 → 320.0 for berberine (IS, $T_R$ 2.2 min) in a positive ionization mode with optimized fragmentor of 120–135 V and collision energy of 25–35 eV, respectively. Concentrations of ceramides in the plasma and protein samples extracted from CD4$^+$ T cell membrane were calculated from the linear regression equation of each ceramide standard curves using the ratio of the peak areas of ceramides and IS.

## T cell differentiation

Naive CD4$^+$ T cells were purified from spleen using CD4$^+$ T cell isolation kit (Miltenyi biotec, 130-104-453) for in vitro and in vivo experiments. To extract protein of cell membrane and cytosol from isolated naive CD4$^+$ T cells, mem-per plus membrane protein extraction kit (Thermo Fisher Scientific, 89848) was used. In some experiments, isolated naive CD4$^+$ T cells were cultured in 6-well or 24-well plates (Costar) in RPMI 1640 medium (Gibco, 11875093) supplemented with 10% FBS (Gibco, 16000-044), 1% P/S (Gibco, 15140122) and 55 µM β−mercaptoethanol (21985-023), and stimulated with plate-bound anti anti-mouse CD3 (2 µg ml$^{-1}$, Invitrogen, 16-0032-85) and anti-mouse CD28 (1 µg ml$^{-1}$, Invitrogen, 16-0281-85). To investigate CD4$^+$ T cell apoptosis by activated- or inactivated-ASM and ceramide, the serum (5%) from WT, Tie2-cre; Smpd1$^{ox/ox}$, or Smpd1$^{-/-}$ mice were added to the naive CD4$^+$ T cells. Apoptotic cells were detected flow cytometry using apoptosis kit (Invitrogen, V13242). To Th17 cell differentiation, naive CD4$^+$ T cells were stimulated in the presence of IL-6 (20 ng ml$^{-1}$), IL-23 (10 ng ml$^{-1}$), IL-1β (10 ng ml$^{-1}$), TGF-β1 (2 ng ml$^{-1}$), anti-IL-4 (10 ng ml$^{-1}$), anti-IFN-γ (10 µg ml$^{-1}$), and anti-IL-2 (10 µg ml$^{-1}$). To Treg cells differentiation, naive CD4$^+$ T cells were stimulated in the presence of TGF-β1 (5 ng ml$^{-1}$). All the cytokines are from Miltenyi biotec. The serum (5 %) from WT, Tie2-cre; Smpd1$^{ox/ox}$, or Smpd1$^{-/-}$ mice were added to the cultures under the Th17 cell or Treg cell-polarizing condition. To confirm the effects of C16-ceramide, recombinant human ASM (rASM), or phosphorlycoline on Th17 cell differentiation, C16-creamide (Avanti, 860516), rASM (Genscript), or phosphorlycholine (MedChemExpress, HY-B2233B) was treated at 10 µM, 2.5 µM, and 10 µM under the Th17 cell-polarizing condition. To examine the inhibitory effects of plasma ASM by ASM antibody (23A12C3, generated for this study) on Th17 cell differentiation, IgG isotype control antibody (50 µg ml$^{-1}$, R&D system, MAB002), ASM antibody (50 µg ml$^{-1}$), and the serum (5%) of Tie2-cre; Smpd1$^{ox/ox}$ was treated to the cultures under the Th17 cell-polarizing condition. After 4 days, cells were collected for flow cytometry, real-time PCR, or western blot.

## Co-cultures of Th17 cells and BV2 microglial cells

Th17 cells were differentiated with or without rASM (2.5 µM, Genscript) and IL17 antibody (30 µg ml$^{-1}$, R&D system, MAB421) as described above. Before 1 day of co-culture with Th17 cells, BV2 microglial cells (Accegen, ABC-TC212S) were seeded in 6-well or 24-well plates (Costar) in RPMI 1640 medium (Gibco, 11875093) supplemented with 10% FBS (Gibco, 16000-044) and 1% P/S (Gibco, 15140122). BV2 microglial cells were co-cultured with Th17 cells for 1 day, and then BV2 microglial cells were collected for real-time PCR, morphology analysis, and Aβ phagocytosis assay. For in vitro Aβ phagocytosis assay, Hilyte Fluor 555-

labeled-Aβ1-42 (AnaSpec, AS-60480) was aggregated for 24 h at 37 °C. Before 1 day of co-culture with Th17 cells, BV2 microglial cells were seeded in 24-well plates (Costar) in DMEM medium (Sigma-Aldrich, D5796) supplemented with 10% FBS and 1% P/S. BV2 microglial cells were co-cultured with Th17 cells for 1 day, and then Aβ (5 µg ml$^{-1}$) was added and incubated for 8 h at 37 °C. For confocal measurement, anti-Iba1 antibody was used to label the cell shape and intracellular Aβ was quantified using MetaMorph software (Molecular Devices).

## Flow cytometry

Immune cells in brain and blood were analyzed by flow cytometry. Single cells from brain were prepared as previously described with minor modifications[56]. Brain was dissected and immediately transferred in ice-cold HBSS (Gibco). After gentle mincing, the brain was digested in a HBSS solution containing collagenase P (0.2 mg ml$^{-1}$, Roche), dispase II (0.8 mg ml$^{-1}$, Roche), DNase I (0.01 mg ml$^{-1}$, Roche), and collagenase A (0.3 mg ml$^{-1}$, Roche) at 37 °C for 1 h under gentle rocking. Digestion was stopped by adding FBS (Gibco) on ice. The supernatants were centrifuged at $250 \times g$ for 10 min at 4 °C. The pellet was resuspended in 25 % BSA (Gibco)/PBS (Gibco) for myelin removal. Following a centrifugation step at $3000 \times g$ for 30 min at 4 °C, the myelin containing supernatant was discarded. The cell pellets were then resuspended in 1 ml of HBSS and filtered through a 40 µm mesh, followed by a washing step in HBSS. The cell pellets were resuspended in 1 ml of red blood cell lysis buffer (BD Biosciences) and incubated at RT for 10 min for lysis of erythrocytes. Subsequently, 2 ml of MACs buffer (Miltenyl Biotec, 130-091-222) was added and centrifuged at $250 \times g$ for 10 min at 4 °C. Blood cells were prepared as previously described with minor modifications[57]. To obtain peripheral blood mononuclear cells (PBMCs), blood was collected in sodium-heparin tube (BD Biosciences, 367871) by cardiac puncture and blood was gently layered in the top of histopaque (Sigma-Aldrich, 10771). After centrifuge ($400 \times g$, 30 min), PBMCs formed in the interphase between histopaque and plasma were collected and washed once with PBS. For analysis of myeloid immune cells, red blood cells were lysed once at 4 °C for 10 min in 0.15 M NH$_4$Cl (STEMCELL Technologies) and washed once with PBS. The cells were stained with the following antibodies for blood and brain macrophage subsets[56,58], monocytes[58,59], neutrophils[58,60], T cells[61], Treg cells, and B cells[62]: mouse anti-CD11b APC (1:100, BD Bioscience, 553312), mouse anti-CD115 PE (1:100, Thermo Fisher Scientific, 12-1152-82), mouse anti-Ly6C FITC (1:100, BD Bioscience, 553104), and mouse anti-Ly6G APC-Cy7 (1:100, BD Bioscience, 557661), mouse anti-Ly6G FITC (1:100, BD Bioscience, 551460), mouse anti-lineage biotin (1:10, Miltenyl Biotec, 130-090-858), anti-biotin streptavidin PB (1:100, Invitrogen, S11222), mouse anti-CD11b PE (1:100, BD Bioscience, 557397), mouse anti-F4/80 APC (1:100, Thermo Fisher Scientific, 14-4801-82), mouse anti-CD45 PerCp Cy5.5 (1:100, BD Bioscience, 550994), mouse anti-CD11b PE (1:100, BD Bioscience, 557397), mouse anti-CD11b PeCy5 (1:100, Tonbo Bioscience, 55-011), mouse anti-CD45 APC-Cy7 (1:100, BD Bioscience, 557659), mouse anti-MHCII PE (1:100, BD Bioscience, 557000), mouse anti-CD206 PE-Cy7 (1:100, Thermo Fisher Scientific, 25-2061-82), mouse anti-CD4 FITC (1:100, BD Bioscience, 553047), mouse anti-CD8 APC (1:100, eBioscience, 17-0081-82), mouse anti-CD25 PE (1:100, eBioscience, 12-0251-82), mouse anti-FoxP3 APC (1:100, eBioscience, 17-5773-82), and mouse anti-B220 PE (1:100, Tonbo Bioscience, 50-0452). For staining of intracellular cytokines, single cells from brain or PBMCs was stimulated with RPMI/10% FBS/1% P/S/ with phorbol myristate acetate (PMA, 50 ng ml$^{-1}$, Sigma-Aldrich, P8139), ionomycin (1 µg ml$^{1}$, Sigma-Aldrich, I0634), Golgi stop (x1500, BD Bioscience, 554724), and Golgi plus (x1000, BD Bioscience, 555029), and incubated for 3 h. Cells were washed and stained with mouse anti-CD4 FITC (1:100, BD Bioscience, 553047) for 30 min. Permeabilization, fixation, and staining of intracellular cytokines with mouse anti-IFNγ APC (1:100,

BD Bioscience, 553047), mouse anti-IL4 PE (1:100, Bio Legend, 504104), and mouse anti-IL17 PerCp Cy5.5 (1:100, BD Bioscience, 553142) were performed with Inside Stain Kit (Miltenyi Biotec, 130-090-477) according to the manufacturer's instructions. Analysis was performed using Attune NxT flow cytometer (Thermo Fisher) and further analyzed using FlowJo analytical software (Tree Star, Inc.). Gating strategy for immune cells in blood, spleen, and brain was shown in Supplementary Figs. 14–16.

### Histological analysis

For immunofluorescence staining, brain was cut on a vibratome (30 μm). Thioflavin S (Sigma-Aldrich, T1892) staining was carried out according to previously described procedures[17,21]. The following antibodies were used: 6E10 (mouse, 1:100, Signet, SIG39300), SMA (mouse, 1:400, Sigma-Aldrich, A2547), Iba1 (rabbit, 1:500, Wako, 019-19941), GFAP (rabbit, 1:500, Dako, N1506), Lamp1 (mouse, 1:200, Abcam, ab24170), and Fibrinogen (Fibrin, rabbit, 1:500, Dako, A008002). All were visualized using Alexa anti-mouse 488 and 633 or Alexa anti-rabbit 488 and 594 as secondary antibodies. To visualize brain microvessels, section was incubated with fluorescein labeled *L. esculentum* lectin (1:100, Vector Laboratories, FL-1171), CD31 (goat, 1:100, R&D system, AF3628), Collagen IV (rabbit, 1:100, Abcam, ab6586), and Aquaporin-4 (chicken, 1:100, Synaptic Systems, 429006). The sections were analyzed with a laser-scanning confocal microscope (FV3000; Olympus) or with a BX51 microscope (Olympus). For quantification of immunostaining of cortex and hippocampus, four images in each region were obtained as shown in Supplementary Fig. 17, and then intensity or cell count was quantified by using "Set color threshold tool" and "Show region statistics tool" of MetaMorph software (Molecular Devices). IMARIS software (Bitplane)[21] was used for analysis of three-dimensional reconstruction of microglia. Confocal images were taken through a z-stack (total z-axis length = 10 μm) and were imported into the IMARIS software. Cell body width was measured, and cell dendrites were automatically detected using the analysis tool. Then, the image was converted into a 3D image, and the cell body volume, process length, number of branches, and terminal tips were automatically quantified. To quantification of amyloid angiopathy stained with Thioflavin S and SMA antibody, four images were taken at 40x magnification within cortex and hippocampus, and the percentage of Aβ intensity within each vessel stained with SMA antibody was measured by MetaMorph software.

### Th17 cell transplantation

To confirm blood-derived Th17 cells in the brain, splenic naive CD4+ T cells, which are EGFP-, were purified from Il17a EGFP mice using a CD4+ T cell isolation kit (Miltenyi biotec, 130-104-453) and cultured in rASM (2.5 μM) under Th17 cell-polarizing conditions (IL-6 (20 ng ml−1), IL-23 (10 ng ml−1), IL-1β (10 ng ml−1), TGF-β1 (2 ng ml−1), anti-IL-4 (10 ng ml−1), anti-IFN-γ (10 μg ml−1), and anti-IL-2 (10 μg ml−1)) for 4 days to generate pathogenic EGFP+ Th17 cells. Three-month-old APP/PS1 mice were intraperitonially injected with IgG isotype control or IL17 antibody (50 μg per mouse, every other day) during 5 weeks of exposure to blood of ASM overexpressing Tie2-cre; Smpd1ox/ox mice. One week before sampling, cultured Th17 cells were intravenously injected (1 × 10^6 cells per mouse) into APP/PS1 mice in parabiosis with Tie2-cre; Smpd1ox/ox mice and treated with IgG isotype control or IL17 antibody. The brain of each mouse was cut on a vibratome (30 μm), and the sections were then stained with GFP (rat, 1:1000, Abcam, ab13970) and collagen IV (rabbit, 1:100, Abcam, ab6586) antibodies for quantification of perivascular and parenchymal EGFP+ Th17 cells. The sections were analyzed with a laser-scanning confocal microscope (FV3000; Olympus). Cell counts were obtained using MetaMorph software (Molecular Devices). Cell numbers per brain region were divided by the respective tissue area and represented as cells/mm².

### Th17 cell staining with BBB

FITC-labeled albumin (Sigma-Aldrich, A9771) was dissolved in buffered saline (10 mg/ml), and the fluorescent dye solution was slowly infused into the tail vein (10 ml/kg) as previously described[63]. With a three-minute interval after the infusion completion, the mice were killed by decapitation and their brains were fixed for 48 h in 4% paraformaldehyde in PBS. Brain was cut on a vibratome (30 μm). Leakage of FITC-labeled albumin into brain parenchyma was observed under a fluorescent microscope. These sections additionally immunoreacted with anti-RORγ antibody (1:100, Rabbit, Abcam, ab207082) determination of Th17 cells in the perivascular and parenchyma of cortex and hippocampus. Cell counts were obtained using MetaMorph software (Molecular Devices). Cell numbers per brain region were divided by the respective tissue area and represented as cells/mm².

### Western blotting

Samples were lysed in RIPA buffer (Cell signaling Technologies, 9806), then subjected to SDS-PAGE and transferred to a nitrocellulose membrane. Membranes were blocked with 5% milk, incubated with primary antibody and then incubated with the appropriate horseradish peroxidase-conjugated secondary antibody[17,21]. Primary antibodies to the following proteins were used: APP (mouse, 1:500, Signet, SIG39300), BACE-1 (mouse, 1:1,000, Millipore, MAB5308), Synaptophysin (rabbit, 1:2000, Abcam, ab32127), PSD95 (mouse, 1:1000, Millipore, MAB1596), Synapsin 1 (rabbit, 1:1000, Synaptic systems, 106 103), MAP2 (chicken, 1:10000, Abcam, ab5392), p-Stat3 (rabbit, 1:1000, Cell Signaling Technology, 9145), Stat3 (mouse, 1:1000, Cell Signaling Technology, 9139), p-JNK (mouse, 1:1000, Cell Signaling Technology, 9255), JNK (rabbit, 1:1000, Cell Signaling Technology, 9252), p-Akt (rabbit, 1:1000, Cell Signaling Technologies, 4060), Akt (rabbit, 1:1000, Cell Signaling Technologies, 4091), p-mTOR (rabbit, 1:1000, Cell Signaling Technology, 5536), mTOR (rabbit, 1:1000, Cell Signaling Technology, 2983), Fibrinogen (Fibrin, rabbit, 1:500, Dako, A008002), Thrombin (goat, 1:100, Santa Cruz Biotechnology, sc-23355), ZO-1 (rabbit, 1:500, Invitrogen, 40-2200), Occludin (mouse, 1:500, Invitrogen, 33-1500), Claudin5 (mouse, 1:500, Invitrogen, 35-2500), and β-actin (1:1,000, Santa Cruz, SC-1615). Rabbit-HRP (1:1000, Cell signaling, 7074), goat-HRP (1:1000, Santa Cruz Biotechnology, sc2020) and mouse-HRP (1:1000, Cell signaling, 7076) were used as secondary antibody. We performed densitometric quantification using the ImageJ software (National Institutes of Health). Images have been cropped for presentation.

### ELISA

For measurement of Aβ40, Aβ42, and IL17, we used commercially available ELISA kits (Invitrogen, KHB3481 for Aβ40; Invitrogen, KHB3441 for Aβ42; and R&D system, M1700 for IL17). Cortex and hippocampus of mice were homogenized in buffer containing 0.02 M guanidine. ELISA was then performed for Aβ40, Aβ42 and IL17 according to the manufacturer's instructions.

### Immunization with ASM peptide and administration of ASM antibody

To examine the possible prophylactic effects of plasma ASM-targeting active immunotherapy, 50 μg/100 μl of ASM protein or 100 μl PBS was mixed with 100 μl of the complete freund's adjuvant (Sigma-Aldrich, F5881) and repeatedly passed through a micro-emulsifying needle (Cadence Science, CAD7977) until the mixture became pasty. Then, 200 μl of mixture was injected intraperitoneally (i.p.) into 6-mo-old WT and APP/PS1 mice. Second and third immunization was performed every 2 weeks with 25 μg/50 μl of ASM protein or 50 μl PBS mixed with 50 μl of the incomplete freund's adjuvant (Sigma-Aldrich, F5506). Four weeks after third immunization, mice were immunized 25 μg/50 μl of ASM protein or 50 μl PBS mixed with 50 μl of the incomplete freund's adjuvant. After 4 weeks, mice were sacrificed for analysis. For passive

immunotherapy, IgG isotype control antibody (50 mg kg$^{-1}$, R&D system, MAB002) or ASM antibody (50 mg kg$^{-1}$, 23A12C3, generated for this study) was injected twice a week i.p. for 8 weeks to 7-mo-old APP/PS1 mice until the age of 9 months.

### In vivo multiphoton microscopy

In vivo multiphoton experiments were performed as previously described[17]. After anesthesia, a cranial window was placed over the partial cortex. Blood plasma was labeled by tail vein injection of TMR-dextran (MW = 40 kD; Invitrogen, D1842). In vivo time-lapse images were acquired at 2, 15, and 30 min after TMR-dextran injection. The leakage from cortical vessels (layer II and III, approximately 100 μm from the cortical surface) was captured in each mouse. Quantification was performed by a blinded investigator by measuring the fluorescent signal intensity in 20 randomly selected 20 μm × 20 μm extravascular areas in brain parenchyma using the NIH ImageJ software integrated density function.

### Behavioral studies

We performed behavioral studies to assess spatial learning and memory in the Morris water maze as previously described[17,21]. Animals were given four trials per day for 10 d to learn the task. At day 11, animals were given a probe trial in which the platform was removed. Fear conditioning was conducted by previously described techniques[17,21]. On the conditioning day, mice were individually placed into the conditioning chamber. After a 60 s exploratory period, a tone (10 kHz, 70 dB) was delivered for 10 s; this served as the conditioned stimulus (CS). The CS co-terminated with the unconditioned stimulus (US), a scrambled electrical footshock (0.3 mA, 1 s). The CS-US pairing was delivered twice at a 20 s intertrial interval. On day 2, each mouse was placed in the fear-conditioning chamber containing the same exact context, but with no administration of a CS or footshock. Freezing was analyzed for 5 min. On day 3, a mouse was placed in a test chamber that was different from the conditioning chamber. After a 60 s exploratory period, the tone was presented for 60 s without the footshock. The rate of freezing response of mice was used to measure the fear memory. All results of behavioral experiments were collected from male mice.

### Recombinant human ASM and monoclonal ASM antibody generation

The synthesis of recombinant human ASM (rASM) and production of mouse monoclonal ASM antibodies were performed by Genscript. For synthesis of rASM, Expi293F cells (Thermo Fisher Scientific, A14527) were transfected with the full-length human ASM cDNA plasmid. The cell culture supernatants were collected on day 6 were used for purification. Cell culture broth was centrifuged. Cell culture supernatant was loaded onto an affinity purification column at an appropriate flow rate. After washing and elution with appropriate buffers, the eluted fractions were pooled and buffer exchanged to the final formulation buffer. The purified protein was analyzed by SDS-PAGE, Western blot analysis to determine the molecular weight (68 kDa) and purity. For production of mouse monoclonal ASM antibodies, screening of 110 hybridomas was performed by ELISA against rASM. Positive clones were expanded and re-tested to confirm epitope reactivity to rASM. Antibody clones were screened through IC$_{50}$ determinations as described below.

### IC$_{50}$ determinations

The fluorescent ASM assay was performed in a 96 well plate using HNPPC (2-N-Hexadecanoyl-4-nitrophenylphosphorylcholine, Toronto Research Chemicals, 60438-73-5) as the substrate. ASM and HNPPC were diluted to 2 μg ml$^{-1}$ and 1 mM in assay buffer (50 mM MES, 0.5 μM ZnCl$_2$, pH 6.5) and incubated for 10 min at 37 °C. ASM antibody (23A12C23) with various concentrations (1 nM to 500 nM) were pre-

incubated for 60 min at 37 °C together with 2 μg ml$^{-1}$ ASM, and then HNPPC was added. For the standard curve, p-nitrophenol was used (Sigma-Aldrich, 241326). After incubation for 6 h at r.t., the reaction was stopped by addition of developing buffer (0.2 M NaOH) and the absorbance of HNPPC was measured at 410 nm. The ASM activity was then calculated: [substrate blank (OD) × conversion factor (pmol/OD, derived using calibration standard p-nitrophenol)/incubation time (h) × amount of enzyme (μg)]. The IC$_{50}$ was analyzed using the GraphPad Prism 7.0 software. Each experiment was performed in triplicate.

### Monoclonal ASM antibody binding assay and ASM titer assay

The recombinant human ASM (2 μg ml$^{-1}$, Genscript) in ELISA coating buffer (Abcam, ab210899) was coated onto MaxiSorp ELISA plates (Thermo Fisher Scientific) overnight at 4 °C. Wells were incubated with blocking buffer (Abcam, ab210904) for 2 h at RT, washed with wash buffer (Abcam, ab206977). ASM antibody (23A12C23) or diluted plasma in blocking buffer were incubated for 2 h at RT. Following washing with wash buffer, Anti-mouse IgG/Biotin (Sigma-Aldrich, B7264) in blocking buffer was added and incubated for 1 h at RT. After standard washing, wells were further incubated with streptavidin HRP (Abcam, ab210901) in blocking buffer for 1 h at RT and results were developed with TMB substrate (Abcam, ab210902). Absorbance was measured at 450 nm using a Varioskan LUX Multimode Microplate Reader (Thermo Scientific). The K$_D$ was analyzed using the SigmaPlot 13.0 software. Each experiment was performed in triplicate. The ASM titers were defined as the dilution factor referring to 50 % of the maximal optical density (ODmax/2).

### Surface plasmon resonance (SPR) spectroscopy

SPR binding experiments were performed using a Biacore® T200 instrument (Biacore, now GE Healthcare). Recombinant ASM was kindly provided by Prof. Edward H. Schuchman (Icahn School of Medicine at Mount Sinai, New York, New York, USA)[64]. ASM was immobilized on the surface of a CM5 sensor chip (GE Healthcare) utilizing standard amine coupling chemistry. The CM5 sensor chip surface was activated by an injection of 0.4 M EDC and 0.1 M NHS at 10 μl min$^{-1}$ for 420 s. HBS-EP buffer containing was used as the running buffer with pH 7.4 (0.01 M HEPES, 0.15 M NaCl, 3 mM EDTA, and 0.005% v/v surfactant P20). ASM (theoretical pI = 6.8) at 25 μg ml$^{-1}$ in 10 mM sodium acetate, pH 4.5, injected over the activated surface at 10 μl min$^{-1}$ for 600 s. The amount of ASM immobilized on the activated surface was typically 5500 response units (RU). The excess hydroxysuccinimidyl groups on the surface were deactivated with 1 M ethanolamine hydrochloride, pH 8.5 for 420 s at a flow rate of 10 μl min$^{-1}$. The surface of a reference flow cell was activated with 0.4 M EDC/0.1 M NHS for 420 s with a flow rate of 10 μl min$^{-1}$, and then deactivated with a 420 s exposure of 1 M ethanolamine at a flow rate of 10 μl min$^{-1}$. With no ligand bound to the flow path, the control flow cell was used to detect nonspecific binding of the small molecules to the sensor chip surface during screening affinity assays. The ASM antibody (23A12C23) were diluted with the assay buffer (10 mM phosphate buffer, 137 mM NaCl, 2.7 mM KCl, pH 7.4, 0.05% v/v surfactant P20) to yield antibody solutions for the assay of concentrations that varied from 0.195 nM to 12.5 nM. Prior to analyte injection, the series S CM5 chip was conditioned with three 30 s cycles of assay buffer followed by three startup cycles, allowing the response to stabilize before analyte injection. Data were collected at a temperature of 25 °C and individual antibody samples were tested from lowest to highest concentrations. During each sample cycle, analyte was injected for 150 s at a flow rate of 10 μl min$^{-1}$. A dissociation period was monitored for 300 s after analyte injection to wash any remaining analyte from the sensor chip before running the next sample. The Biacore T200 was programmed to run an automated assay with the various antibody samples. The responses measured in

the blank flow cell (control) were subtracted from the response measured in the flow cell with protein immobilized. Equilibrium constants (KD) were calculated using the 'kinetic' model in Biacore T200 Evaluation Software. All experiments were repeated three times.

### RNA isolation and real-time PCR analysis

RNA was extracted from the brain homogenates and cell lysates using the RNeasy Lipid Tissue Mini kit and RNeasy Plus Mini kit (QIAGEN) according to the manufacturer's instructions. cDNA was synthesized from 5 μg of total RNA using a commercially available kit (Takara Bio Inc.). Quantitative real-time PCR was performed using a Corbett research RG-6000 real-time PCR instrument. Used primers are described in Supplementary Table 2.

### Statistical analysis

Sample sizes were determined by G-Power software (ver 3.1.9.4, with $\alpha = 0.05$ and power of 0.8). In general, statistical methods were not used to re-calculate or predetermine sample sizes. Variance was similar within comparable experimental groups. Individuals performing the experiments were blinded to the identity of experimental groups until the end of data collection and analysis for at least one of the independent experiments. All data are representative of at least three independent experiments. Comparisons between two groups were performed with a two-tailed student's t test. In cases where more than two groups were compared to each other, one way analysis of variance (ANOVA) was used, followed by Tukey's HSD test. All statistical analyses were performed using GraphPad Prism 7.0 software. $P < 0.05$ were considered as statistically significant.

### Reporting summary

Further information on research design is available in the Nature Portfolio Reporting Summary linked to this article.

## Data availability

All data supporting the findings of this study are available in the article and its Supplementary Information. The datasets generated and/or analyzed during the current study are also available from the corresponding authors upon reasonable request. Source data are provided with this paper.

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

## Acknowledgements

We particularly would like to thank Yu Sin Han, Eun Yeong Lim, and Yun Ju Park for technical assistance. This work was supported by the National Research Foundation of Korea (NRF) grant funded by the Korea government (MSIT) (2018M3C7A1056513 to H.K.J, 2020R1A2C3006875 to J.S.B, 2020R1A2C3006734 to H.K.J, 2020R1A4A2002691 to H.K.J). This research was also supported by a grant of the Korea Health Technology R&D Project through the Korea Health Industry Development Institute (KHIDI), funded by the Ministry of Health & Welfare and MSIT, Republic of Korea (HU20C0345 to J.S.B).

## Author contributions

B.J.C. and M.H.P. designed and performed experiments and wrote the paper. K.H.P., W.H.H., H.J.Y., H.Y.J., J.Y.H., M.R.C., and K.Y.K. performed experiments and analyzed data. J.L., I.S.S., M.P., and M.K.C. performed LC-MS/MS experiments and analyzed data. E.G., M.R. and J.K. generated and provided *Smpd1* ox/ox mice. S.H.K. performed normal and AD patient plasma experiment. E.H.S. provided the *Smpd1*–/– mouse and recombinant ASM. C.W.H., C.K., S.H.K., E.H.S., H.K.J., and J.S.B. interpreted the data and reviewed the paper. H.K.J. and J.S.B. designed the study and wrote the paper. All authors discussed results and commented on the paper.

## Competing interests

The authors declare no competing interests.
