## [Peer Review File · Nature Communications]

Immunotherapy targeting plasma ASM is protective in a mouse model of Alzheimer's diseaseEditorial Note: Parts of this Peer Review File have been redacted as indicated to remove third-party material where no permission to publish could be obtained.

REVIEWER COMMENTS

Reviewer #1 (Remarks to the Author):

Overall, this is an interesting manuscript, with potential clinical implications, showing that ASM is a negative player in Alzheimer's disease, and that neutralizing it can provide a potential therapy.

General comments:

The proposed mechanism is not fully substantiated. (see the comment to Figure 6)

The abbreviations used in the figures are very confusing.

Iba1 is not an exclusive marker for microglia.

In most figures, no indications are given for the number of repetitions.

Specific comments

Figure 1, the authors concluded (line 120) that the source of ASM is from the BBB-ECs, but no direct evidence is provided to support this contention.

Throughout the study, and especially in Figure 2, the authors neglected to include a key control parabiosis of WT-WT and parabiosis of WT-WO-ASM mice. There is no evidence for the effect of ASM overexpression on the hetero-parabiosis mice in terms of ASM levels, Th17, and cognitive performance.

Figure 3 lacks a control of neutralizing L-17 in APP mice (AD mice), with no parabiosis

Figure 6 is a key figure, and is missing:

1. Measurements of Th17 levels with and without vaccination, to confirm the proposed connection between ASM, effect on the immune system, and Th17.
2. Immunization of wild type mice with ASM is missing.
3. Morris Water Maze, the learning curve of the WT animals is very poor, although the overall differences after long period are significant.

Reviewer #2 (Remarks to the Author):

In this study Choi et al. investigate the molecular and cellular mechanisms downstream of elevated plasma ASM in an APP/PS1 mouse model of Alzheimer's disease. This builds on prior work from the same group on the role of ASM in brain aging and neurodegeneration. The authors describe an increase in plasma ASM in human MCI/AD and in 9 month old, but not 3 or 6 month old, APP/PS1 compared to control mice. To selectively investigate the role of systemic ASM, the authors combine a parabiosis approach with a transgenic mouse model overexpressing ASM in endothelial cells (ASM OE) leading to increased plasma ASM level. When comparing young APP mice parabiosed to ASM OE mice to those attached to WT mice, Choi et al. observe accelerated neurodegenerative phenotypes including increased Amyloid-beta pathology and microglial/astrocyte activation. Using a series of in vitro and in vivo experiments, the authors demonstrate that elevated ASM levels lead to several changes in immune cell populations including an increase in TH17 cells and a decrease in Tregs. These changes were observed in both peripherally in the blood and spleen as well as centrally in the brain. The authors go on to show that anti-IL17 treatment, which ablates TH17 cells, reduces pathological phenotypes in APP/PS1 mice parabiosed to ASM-OE mice. Lastly, using a series of active and passive anti-ASM immunization approaches, the authors show that blocking systemic ASM in both the ASM-OE conditions, and more importantly also in normal 7-9 month old APP/PS1 mice, reduces peripheral and central immune cell shifts, as well as Amyloid-beta pathology, glial activation, synaptic changes and behavioral impairments. This is a timely and compelling study on the systemic plasma factor ASM and its role on immune cell function, blood-brain barrier integrity and

neurodegenerative changes in the brain. While the findings are of high quality and relevance, the experimental approaches are, at times, unnecessarily complicated and some of the data and conclusions can also be interpreted differently. Below, is a series of questions for the authors to address.

Major

1. It is somewhat surprising to see which experiments/data Choi et al. decided to feature in the main figures versus the supplement. At times, the reader is almost obliged to go through the extended figures, since they feature crucial data and experiments. For example, Figure 5 features a convoluted parabiosis experiment, in which a *Smpd1* knockout mouse is used to generate anti-bodies against ASM to block plasma ASM in its surgically connected APP/PS parabiotic partner. In Figure 6, the authors then go on use a much more controlled active immunization paradigm, that comes to the same conclusions, raising the question whether the data in Figure 5 is necessary at all. Conversely, in Extended Figure 10, the authors use passive immunization with a newly developed neutralizing antibody, which is arguably a much more therapeutically relevant approach. Importantly, both Figure 6 and Extended Figure 10 include WT baseline conditions which allow for a better assessment of the extent of the rescue, while Figure 5 does not. Moreover, full body *Smpd1* knockout mice in Figure 5 display additional changes such as large difference in neutrophils (Figure 5j,o), which further confounds the interpretation of this experiment. Given these confounds and limitations, it is unclear to us why Figure 5 was included in this manuscript. Correspondingly, we recommend the authors feature the Extended Figure 10 in the main manuscript in lieu of Figure 5.

2. The authors show that ASM is elevated in 9 but not 6 months old APP/PS1 mice (Extended Figure 1c). Similarly, the authors show that TH17 cells are increase in 9 months old APP/PS1 mice, although it is unclear if these changes are present at 3 or 6 months (Extended Figure 8b-g). While the authors used anti IL17 treatment in the accelerated ASM-OE parabiosis condition (Figure 3+4), it would be more informative to assess the benefits of this treatment in 6-9 months old APP/PS1 in the absence of ASM overexpression.

3. The authors lay out a narrative that ASM alters CD4 cell differentiation, which leads to an increase of TH17 cells. These peripheral TH17 cells then impair BBB function, migrate into the brain, where they drive neuroinflammatory changes, protein aggregation and functional impairments. While the authors provide some evidence supporting this narrative, a lot of their data can also have alternative interpretations and several aspects of this mechanism have not actually been definitively demonstrated. For example, while Choi et al show an increase in TH17 cells in the brain, they do not show that these cells are systemically derived from the blood. This could for example be shown using GFP labeled cells. Moreover, the increase/decrease in CNS TH17 cells is always paralleled by an increase/decrease in peripheral TH17 cells (Figure 3b,h +Extended Figure 3). Thus, the authors cannot experimentally distinguish between the effect of peripheral versus brain TH17 cells. While it is convenient to speculate that brain TH17 cells mediate all the effects, peripheral TH17 could also impair brain function by creating a more inflammatory systemic milieu. As such, several peripheral pro-inflammatory chemokines and cytokines have previously been shown to be able to enter the brain and negatively regulate brain function. While some of these questions may be challenging to experimentally address, the authors should at the least discuss these limitations and provide plausible alternative interpretations of their data in the discussion.

4. Immunostainings and quantifications. The authors should provide a more thorough characterization and description of several immunostainings and analyses. First, infiltrating brain immune cells were mostly analyzed by FACS of dissociated whole brain, thus obfuscating specific regional origins of the cells. While some immunostaining of cortex and hippocampus were performed, the authors should provide a more careful quantification of the infiltrating immune cells and distinguish between, vascular, perivascular and parenchymal localization. Additionally, Lectin labelling was predominantly used as a marker for the vasculature. However, Lectin does not necessarily label all blood vessels and additionally may also label reactive glial cells in AD mouse models. The authors should consider complementing their vascular analysis with additional vascular markers (e.g., CD31, CollagenIV or Aqp4).

Second, it is often unclear which exact cortical/hippocampal regions were quantified. The inclusion of a nuclear DAPI stain in representative images, a schematic depicting the quantified region or a more extensive description in the figure legend should be included. Additionally, several quantifications lack representative images. For example, in Extended Fig 7d/9i, the authors provide quantification of CAA. However, most mouse models of AD do not recapitulate CAA very robustly. Moreover, the authors do not provide any methodology of how CAA was assessed.

Minor

1. The authors should specify the mouse ages more clearly in the figure legends. This especially applies to the extended figures which, unlike the main figures, do not feature schematic with the experimental paradigms.
2. Figure 2. At what age was the serum collected that was used in the in vitro experiments? The authors should provide more experimental detail regarding plasma/serum collection methods including centrifugation speeds etc. Moreover, are the CD4 T cells derived from WT or from APP transgenic mice and at what age were they collected? Additionally, the authors appear to use an immortalized microglial cell line, BV2, when referring to the microglia experiments in vitro. However, BV2 cells are not microglia, which should be stated more clearly in the manuscript.
3. What ASM peptide was used for the active immunization?
4. Several Western blots have very high background and it is unclear how the authors were able to quantify individual bands (eg Figure 4h)
5. Figure 6 f: Representative in vivo time-lapse image versus the quantification show a 30min versus a 20min time-point.
6. Do the authors have a possible explanation for the massive increase in peripheral neutrophils in the periphery (Figure 5j) and decrease in brain neutrophils (Figure 5o) in response to Smpd1 KO parabiosis?

Reviewer #3 (Remarks to the Author):

In this study, Choi et al described the pathogenic Th17 cell differentiation from CD4+ T cells and the acceleration of AD brain pathology and neurobiological behavioral impairments (i.e., spatial learning and memory decline in the Morris water maze) by increased ASM activity. This study also demonstrates the prevention of these phenotypes by Smpd1 deletion and/or SAM-related active/passive immunotherapy. Overall, this is a very interesting and nicely conducted study that provides new insights into the cellular and molecular regulation of Th17 cell differentiation by elevated ASM and the consequent effects on AD pathogenesis. While I am enthusiastic about this novel and interesting study, there are a few areas in which this manuscript could be improved.

Major comments:

1. Are all effects of SAM observed in this manuscript mediated by ceramides? While ASM activity in the plasma of AD patients and mouse models increases, no changes in the levels of plasma ceramides were detected (Extended Data Fig. 1a-d). More importantly, the APP/PS1-Smpd1ox/ox parabiotic mice show increased plasma ASM activity compared to APP/PS1-WT parabiotic mice (Fig. 1b), whereas plasma ceramide levels are not significantly different in these mice (Fig. 1c). Clear discussion and explanations on how the “disassociation” between elevated plasma ASM activity and unchanged plasma ceramide level affects the observed protection against AD phenotypes in mice by the immunotherapy targeting plasma ASM will improve the manuscript.

2. Fig. 2e, g, how does elevated plasma ASM in Smpd1ox/ox serum lead to elevated ASM activity in CD4+ T cell membranes? Is the plasma ASM recruited to the cell membrane of CD4+ T cells or internalized and trafficked to vesicles or the lysosome of these cells? Is the membrane ASM of CD4+ T cells activated by some factors presented in Smpd1ox/ox serum but absent in Smpd1-/- serum? Are there other reasons? Please discuss and explain.

3. Page 11, lines 201-204, "Collectively, these results (Fig. 2) suggested that elevated plasma ASM leads to elevated ASM activity and ceramide in CD4+ T cell membranes (Fig. 2e-h), resulting in apoptosis and pathogenic Th17 cell differentiation by stimulating downstream signals such as, Stat3, JNK, AKT, and mTOR (Fig. 2i-l)." Based on these data and this summary, is the conclusion that the membrane ASM and membrane ceramide are responsible for the observed effects of elevated ASM activity in AD patients and mouse models correct? If yes, the emphasize of the plasma ASM and the immunotherapy targeting plasma ASM in this manuscript might not be the best way to tone the study and the conclusion. If such a conclusion is not the conclusion of the authors, what conclusion should be drawn on this issue?

4. Indeed, it has been found that vesicles containing ASM can translocate to the plasma membrane and release ASM to the outer leaflet or to the extracellular space. This ASM species is referred to as secretory ASM (Rotolo et al., 2005), lysosomal ASM (Li et al., 2012; Defour et al., 2014), or ASM without specified affiliation (Grassmé et al., 2001; Verdurmen et al., 2010). Regardless, the term of "the plasma ASM" should be used to indicate the ASM in the fluid part of blood but it should not be also used for the membrane ASM of CD4+ T cells or other cells.

Minor comments:

1. In Fig. 1i, the representative immunofluorescence images and quantification of astrocytes (i, GFAP) appear not to match. Why GFAP staining shows such a remarkable "region specific" in the cortex? If reactive astrocytes around A β plaques partially contribute to such an uneven distribution of astrocytes in the cortex brain region, why such a pattern is not seen in microglia distribution in the same brain region (h, Iba1)? Double-labelling of A β using 6E10 antibody and GFAP or Iba1 will be better.

2. In Fig. 2l, does SAM treatment cause similar activation of these pathways as did by C-16 ceramide? Is there any possible involvement of the other hydrolysate of sphingomyelin, phosphorylcholine in the effects of elevated ASM activity in this manuscript?

3. Multi-dimensional views of microglial phagocytosis of Fluor 555-labelled A β in Fig. 1i and Fig. 3m that include image information in the x, y, and z dimensions are better presentation of phagocytic function.

4. In Fig. 3k-m, IgG isotype/vehicle control, as did in Fig. 3a-j, will be better than no treatment control.

5. Overall this manuscript is well-written. However, some result description should be more accurate. For examples, "Co-culture of microglia with the ASM treated Th17 cells revealed a deficient Fluor 555-labeled A β phagocytic capacity of the microglia, while Th17 cells treated with ASM or IL17 antibody did not exhibit this abnormal microglia function(Fig. 3m). (lines 236 & 237). The inaccurate writing in the clause could cause confusion or misunderstanding. How Th17 cells could exhibit microglia function? Should the clause be "while microglia co-cultured with ASM-primed Th17 cells treated with IL17 antibody did not exhibit this abnormal function"?

6. Lines 346-349, "These data revealed the significant positive effects of immunization of APP/PS1 mice with an ASM peptide on various pathological features, including changes of immune cell populations, BBB damage, neuroinflammation, A β accumulation, synapse loss, and learning and memory impairment". Lines 44-48, "Antibody-based immunotherapy targeting plasma ASM showed efficient inhibition of ASM activity in the blood of AD mice and, interestingly, this treatment led to positive effects on blood-brain barrier (BBB) disruption, neuroinflammation, A β deposits, synapse loss, and memory impairment by suppressing pathogenic Th17 cells". My suggestion is to replace the

“positive effects” in these two sentences with prophylactic effects to make them clearer.

Reviewer #1 (Remarks to the Author):

Overall, this is an interesting manuscript, with potential clinical implications, showing that ASM is a negative player in Alzheimer's disease, and that neutralizing it can provide a potential therapy.

We would like to thank the reviewer for these comments.

General comments:

The proposed mechanism is not fully substantiated. (see the comment to Figure 6)

Please see our answer below in response to comments about Figure 6.

The abbreviations used in the figures are very confusing.

We apologize for the confusion and have carefully listed the abbreviations in the revised manuscript (**Fig. 1,3,4 and Extended Data Fig. 2-8**).

Iba1 is not an exclusive marker for microglia.

We would like to thank the reviewer for this comment and agree that Iba-1 is not an exclusive marker for microglia. However, to better understand why we used Iba-1 as a microglial marker in this study, we would like to explain below.

Iba-1 is a cytoplasmic helix-loop-helix protein with F-actin binding and actin-cross-linking activity involved in cell motility and phagocytosis (*Biochem. Biophys. Res. Commun.* 2001. 286:292–297; *J Neurosci.* 2011. 31:6277–6288; *Cereb Cortex.* 2012. 22:1442–1454; *Brain Behav Immun.* 2014. 37:1-14). It is expressed within the cytoplasm, including in the slender protrusions of both ramified and activated microglia, and is upregulated during the reactive response. Many studies suggest that Iba-1 is a suitable marker of microglia activation or phagocytosis in the brain of neurodegenerative diseases including AD, particularly when used together with analysis of microglial morphology. Previous work, including our own, have also detailed the morphology of microglia using Iba-1 staining and further analyzed quantitative morphology using three-dimensional reconstruction (*Cell Death Dis.* 2012. 3:e379; *Sci Rep.* 2012. 2:809; *Nat Neurosci.* 2013. 16:1618-1626; *J Neuroinflammation.* 2014. 11:182; *Glia.* 2016. 64:1285-1297; *PNAS.* 2019. 116:23426-23436; *Nature comm.* 2018. 9:1479;

Nature comm. 2020. 11:2358; *PNAS.* 2022. 119:e2115082119).

Tmem119 and P2ry12 are known to be exclusive markers for microglia, and have been reported to provide a homeostatic, functional signature of the cells. During the course of neurodegeneration, microglia lose their homeostatic molecular signature (*Acta Neuropathol. Commun.* 2015. 3:31; *Cell.* 2017. 169:1276; *Immunity.* 2017. 47: 566) and show decreased expression of Tmem119 and P2ry12 in the AD brain (**Additional Fig. 1**). We also confirmed that expression of these marker was decreased around A β in the cortex of APP/PS1 mouse compared to WT mouse, while Iba-1 expression increased around A β (**Additional Fig. 2**). These data indicate that Tmem119 and P2ry12 may be not suitable to confirm microglia activation in AD brain.

Based on previous studies, in the current manuscript we therefore confirmed microglial activation and studied the morphology of the microglia via Iba-1 staining. Although Iba-1 is sometimes not an exclusive marker for microglia, we believe that Iba-1 is the most appropriate marker for confirming microglial activation and phagocytosis morphologically in the AD brain.

J Neuroinflammation. 2018. 15:274

eLife. 2020. 9:e54083.

Int. J. Mol. Sci. 2020. 1:678

PNAS. 2019. 116:23426-23436.

Additional Fig. 1. References of Tmem119 and P2ry12 expression in the AD brain. The expression of Tmem119 and P2ry12 decreased in the AD brain (*J Neuroinflammation*. 2018. 15:274; *PNAS*. 2019.116:23426-23436; *Int. J. Mol. Sci*. 2020. 1:678; *eLife*. 2020. 9:e54083).

Additional Fig. 2. The representative images of Tmem119 (cyan, rabbit, Abcam, ab209064), Iba-1 (red, goat, Novus, NB100-1028), and P2ry12 (cyan, rat, Biolegend, 848002) expression around A β (green, ThioS) in the cortex of 9-month-old WT and APP/PS1 mouse. Scale bar = 20 μ m.

In most figures, no indications are given for the number of repetitions.

According to the reviewer's comment, we have added the number of repetitions in each figure legend of the revised manuscript.

Specific comments

Figure 1, the authors concluded (line 120) that the source of ASM is from the BBB-ECs, but no direct evidence is provided to support this contention.

First of all, we apologize to the reviewer for any confusion due to the lack of support data about the correlation between the increased plasma ASM and BBB-ECs ASM in AD mice.

Our previous studies have shown that the age-related increase of plasma ASM activity may be derived from the BBB-ECs (*Neuron*. 2018. 100:167-182; *Exp. Mol. Med.* 2020. 52:380-389). Based on these reports, we speculated in this manuscript that the increase of ASM activity in the BBB-ECs of APP/PS1 mice led to elevation of ASM activity in the plasma.

To clarify this hypothesis, we performed additional experiments. For specific inhibition of ASM activity in the BBB-ECs, control or *Smpd1* miR RNAi mixed with *in vivo-jetPEI*TM solution (Polyplus-transfection, 201-50) were rapidly injected into the tail vein twice a week for 3 weeks in 8-month-old APP/PS1 mice. This procedure was previously shown to achieve selective gene transfer to brain endothelial cells (*Nat Med.* 2012. 18:1658-1664; *Nat Commun.* 2012. 3:849; *J Exp Med.* 2015. 212:2267-2287). Our previous study also showed that *Smpd1* miR RNAi with human Endoglin promoter can specifically decrease ASM activity in brain microvessels (*Neuron*. 2018. 100:167-182). After 3 weeks, we found a decrease of ASM activity in the brain microvessels and plasma of *Smpd1* miR RNAi-treated APP/PS1 mice compared with control miR RNAi-treated mice (**Additional Fig. 3**). Although further studies may be needed to definitely prove that the plasma ASM is derived from BBB-ECs, these data indicate that BBB-ECs may be the source.

However, despite this new data because of the potential uncertainty we have removed this sentence and the extended data in Fig. 1e from the original manuscript to avoid reviewer and reader confusion.

Additional Fig. 3. ASM activity in the brain microvessels and plasma of control or *Smpd1* miR RNAi-treated APP/PS1 mice (n = 5-6 per group). Student's t test. $**P < 0.01$. All error bars indicate s.e.m. All data analysis was done at 9-mo-old mice.

Throughout the study, and especially in Figure 2, the authors neglected to include a key control parabiosis of WT-WT and parabiosis of WT-WO-ASM mice. There is no evidence for the effect of ASM overexpression on the hetero-parabiosis mice in terms of ASM levels, Th17, and cognitive performance.

As recommended by the reviewer, we performed new experiments that include the control parabiosis of WT-WT pair and OX-WT pair as shown in **Fig. 1a**.

Fig. 1b indicates that the OX-WT and OX-APP/PS1 parabiotic mice showed increased levels of plasma ASM activity compared to WT-WT and WT-APP/PS1 parabiotic mice at 5 weeks post-surgery. No significant differences in plasma ceramide levels were found in these mice (**Fig. 1c**). To determine whether the increase of plasma ASM activity was accompanied by local changes within the brain, we measured ASM activity in the cortex and hippocampus of these mice. Contrary to the plasma results, there was no difference in ASM activity in the cortex and hippocampus (**Fig. 1d and Extended Data Fig. 2a**), indicating a specific increase of plasma ASM activity in these young WT and APP/PS1 mice that had been joined by parabiosis with ASM overexpressing (OX) mice.

Next, we investigated the changes of neuropathological features in the brain of each parabiotic mice. Three-month-old of APP/PS1 mice exchanged by parabiosis with blood of OX mice showed early A β accumulation by thioflavin S (ThioS) and 6E10 staining, as well as by A β 40 and A β 42 ELISA in the cortex and hippocampus (**Fig. 1e-g and Extended Data Fig. 2b-d**). Additionally, these mice exhibited highly activated microglia and astrocytes. Although WT mice exchanged by parabiosis with blood of OX mice also showed activated microglia and astrocytes and a pro-inflammatory microglia phenotype in the cortex and hippocampus, this was less than in APP/PS1 mice exchanged by parabiosis with blood of OX mice (**Fig. 1h-k and Extended Data Figures 2e-h**). These data suggested that overexpression of plasma ASM activity contributed to acceleration of A β accumulation and neuroinflammation in the brain of young APP/PS1 mice despite the fact that the brain tissue ASM activity was not elevated.

Considering that overexpressed plasma ASM led to acceleration of neuroinflammation in the brain of young WT and APP/PS1 mice, we investigated the effect of plasma ASM on

immune cells. There were no significant differences in the changes of leukocytes subpopulations such as neutrophils, monocytes, macrophages, CD8⁺ T cells, and B220⁺ B cells between these groups (**Extended Data Fig. 3a,b**). However, we found a decrease of CD4⁺ T and Treg cells and an increase of Th17 cells in the blood of WT and APP/PS1 mice exchanged by parabiosis with blood of OX mice compared to parabionts of WT-WT and WT-APP/PS1 mice. These changes were greater in the blood of APP/PS1 mice exchanged by parabiosis with blood of OX mice than WT mice exposed to OX blood (**Extended Data Fig. 3c**). The results in the spleen and brain tissue also showed similar changes in the CD4⁺ T, Th17, and Treg cells (**Extended Data Fig. 3d-g**). These findings suggested that blood-derived Th17 cells might be involved in the microglia-mediated neuroinflammation and early A β accumulation of young APP/PS1 mice exposed to ASM overexpressing blood.

In the revised manuscript, we have included the results regarding the control parabiosis pairs in **Fig. 1** and **Extended Data Fig. 2,3** with data analyses and interpretations.

Fig. 1. Young APP/PS1 mice in parabiosis with *Smpd1*^{ox/ox} mice exhibit acceleration of A β deposits and neuroinflammation in the cortex. a, Schematic showing parabiotic pairings. b,

ASM activity in plasma (n = 8-14 per each group). **c**, Ceramide concentration in plasma (n = 5-6 per each group). **d**, ASM activity in the cortex (n = 5-7 per each group). **e**, Left, representative immunofluorescence images of thioflavin S (ThioS, A β plaques). Scale bars, 50 μ m. Right, quantification of area occupied by A β plaques (n = 5-7 per each group). **f**, Quantification of 6E10 (n = 5-7 per each group). **g**, Analysis of A β 40 and A β 42 depositions in cortex using ELISA kits (n = 5-6 per each group). **h and i**, Immunofluorescence images and quantification of microglia (h, Iba-1, red) and astrocyte (i, GFAP, red) with A β plaques (ThioS, green) (n = 5-7 per each group). Scale bars, 50 μ m. **j**, mRNA levels of inflammatory markers (n = 4-6 per each group). Pro-inflammatory marker: *Tnf- α* , *Il-1 β* , *Il-6*, and *iNos*, Immunoregulatory cytokine: *Il-10*, Anti-inflammatory marker: *Il-4*, *Tgf- β* , and *Arg1*. **k**, Left, imaris-based three-dimensional images (Scale bars, 10 μ m) of microglia. Right, imaris-based automated quantification of microglial morphology (n = 4 mice per each group). **l**, Immunofluorescence images of ThioS (A β plaques, green) encapsulated within Lamp1⁺ structures (phagolysosomes, blue) in microglia (Iba1, red) present in cortex of APP/PS1 mouse in parabiosis with *Smpd1*^{ox/ox} mice. Scale bars, 30 μ m. AD, APP/PS1; OX, *Smpd1*^{ox/ox} (Tie2-cre; *Smpd1*^{ox/ox} mice). b-k, One-way analysis of variance, Tukey's post hoc test. *P < 0.05, **P < 0.01, ***P < 0.001, ****P < 0.0001. The data were collected from 3 independent experiments. All error bars indicate s.e.m. All data analysis was done at 4.5-mo-old mice.

Extended Data Fig. 2. Young APP/PS1 mice in parabiosis with *Smpd1*^{ox/ox} mice show acceleration of Aβ deposits and microglia-mediated pro-inflammation in the hippocampus. **a**, ASM activity in the hippocampus (n = 5-7 per each group). **b**, Left, representative immunofluorescence images of thioflavin S (ThioS, Ab plaques). Scale bars, 50 μm. Right, quantification of area occupied by Aβ plaques (n = 5-6 per each group). **c**, Quantification of 6E10 (n = 5-7 per each group). **d**, Analysis of Aβ40 and Aβ42 depositions using ELISA kits (n = 5-6 per each group). **e and f**, Immunofluorescence images and quantification of microglia (e, Iba-1, red) and astrocyte (f, GFAP, red) with Aβ plaques (ThioS, green) (n = 5-7 per each group). Scale bars, 50 μm. **g**, mRNA levels of inflammatory markers in cortex (n = 4-6 per each group). Pro-inflammatory marker: *Tnf-α*, *Il-1β*, *Il-6*, and *iNos*, Immunoregulatory cytokine: *Il-10*, Anti-inflammatory marker: *Il-4*, *Tgf-β*, and *Arg1*. **h**, Left, imaris-based three-dimensional images (Scale bars, 10 μm) of microglia. Right, imaris-based automated quantification of microglial morphology (n = 4 per each group). **i**, Western blot analysis and quantification of APP and β-CTF level and BACE1 level in the cortex and hippocampus (n = 4-5 per each group). AD, APP/PS1; OX, *Smpd1*^{ox/ox} (Tie2-cre; *Smpd1*^{ox/ox} mice). a-h, One-way analysis of variance, Tukey's post hoc test. i, Student's t test. *P < 0.05, **P < 0.01, ***P < 0.001, ****P < 0.0001. The data were collected from 3 independent experiments. All error bars indicate s.e.m. All data analysis was done at 4.5-mo-old mice.

Extended Data Fig. 3. Parabionts of young APP/PS1-*Smpd1*^{ox/ox} mice exhibit reduction of CD4⁺ T cells and Treg cells and increase of Th17 cells in the blood and brain. **a**, Percentage of neutrophils (CD11b⁺ Ly6C⁺ CD115⁻), pro (CD11b⁺ Ly6C⁺ CD115⁺)- or anti (CD11b⁺ Ly6C⁻ CD115⁺)-monocytes, and macrophages (Lin⁻ CD11b⁺ F4/80^{hi} Ly6C⁻) in the blood (n = 5 per each group). **b**, Percentage of CD4⁺, CD8⁺ T cells, and B220⁺ B cells in the blood (n = 4-6 per each group). **c**, Percentage of Th1 (CD4⁺ IFN- γ ⁺), Th2 (CD4⁺ IL4⁺), Th17 (CD4⁺ IL17⁺), and Treg (CD4⁺ CD25⁺ FOXP3⁺) cells in the blood (n = 5-6 per each group). **d**, Percentage of CD4⁺ T cells, Th17 (CD4⁺ IL17⁺), and Treg (CD4⁺ CD25⁺ FOXP3⁺) cells in the spleen (n = 5 per each group). **e**, Percentage of neutrophils (CD45^{hi} CD11b⁺ Gr-1⁺), monocytes (CD45^{hi} CD11b⁺ Ly6C^{hi} Ly6G^{low}), and M1 (CD45^{hi} CD11b⁺ Ly6G⁻ F4/80⁺ MHCII^{hi})- or M2 (CD45^{hi} CD11b⁺ Ly6G⁻ F4/80⁺ CD206⁺)-macrophages in the brain (n = 4-5 per each group). **f**, Percentage of CD4⁺, CD8⁺ T cells, and B220⁺ B cells in the brain (n = 4 per each group). **g**, Percentage of Th1 (CD4⁺ IFN- γ ⁺), Th2 (CD4⁺ IL4⁺), Th17 (CD4⁺ IL17⁺), and Treg (CD4⁺ CD25⁺ FOXP3⁺) cells in the brain (n = 4-5 per each group). All immune cells were analyzed by flow cytometry. AD, APP/PS1; OX, *Smpd1*^{ox/ox} (Tie2-cre; *Smpd1*^{ox/ox} mice). a-g, One-way analysis of variance, Tukey's post hoc test. *P < 0.05, **P < 0.01, ***P < 0.001. The data were collected from 3 independent experiments. All error bars indicate s.e.m. All data analysis was done at 4.5-month mice.

Figure 3 lacks a control of neutralizing L-17 in APP mice (AD mice), with no parabiosis

To help the reviewer better understand why we did not include IL17 antibody-treated APP/PS1 mice group with no parabiosis in Fig. 3, we would like to explain below.

Three-month-old APP/PS1 mice used in Fig. 3 showed no significant difference in blood ASM activity, A β deposit and neuroinflammation compared to age-matched WT mice (**Fig. 1 and Extended Data Fig. 1,2**). Moreover, these mice did not have increased Th17 cells in blood, spleen, and brain (**Extended Data Fig. 3c,d,g**), indicating that an IL17 antibody would not result in any specific changes in young APP/PS1 mice with no parabiosis.

To confirm this, we have performed additional experiments including control or IL17 antibody-treated 3-month-old APP/PS1 group with no parabiosis as shown **Fig. 3a**. Although IL17 antibody injection induced reduction of Th17 cells in the spleen and blood of young APP/PS1 mice with no parabiosis compared to a control antibody injected group, A β accumulation was not changed between the groups (**Fig. 3b,c**). On the other hand, IL17 antibody injection in the young APP/PS1 mice exposed to blood of OX mice prevented early A β accumulation (**Fig. 3c**). These data indicated that IL17 antibody prevents early A β accumulation in young APP/PS1 mice by inhibiting Th17 cells only under conditions where plasma ASM is overexpressed.

As suggested by the reviewer, we have added the data of control of IL17 antibody-treated APP/PS1 mice group with no parabiosis in **Fig. 3a-c** of the revised manuscript.

Fig. 3a-c. Inhibition of blood-derived pathogenic Th17 cells prevents A β deposits in young APP/PS1 mice exposed to blood of Tie2-cre; *Smpd1*^{ox/ox} mice. **a**, Scheme of experimental procedures. IgG isotype control or IL17 antibody was injected (50 μ g/mouse, i.p. every other day) into 3-month-old of APP/PS1 mice in parabiosis with *Smpd1*^{ox/ox} mice for 5 weeks during parabiosis or no blood exchange. **b**, Percentage of Th17 cells in the spleen and blood using flow cytometry (n = 4-6 per each group). **c**, Quantification of area occupied by A β plaques (ThioS) (n = 4 per each group). AD, APP/PS1; OX, *Smpd1*^{ox/ox} (Tie2-cre; *Smpd1*^{ox/ox} mice). One-way analysis of variance, Tukey's post hoc test. *P < 0.05, **P < 0.01, ***P < 0.001. The data were collected from 3 independent experiments. All error bars indicate s.e.m. All data analysis was done at 4.5-mo-old mice.

Figure 6 is a key figure, and is missing:

1. Measurements of Th17 levels with and without vaccination, to confirm the proposed connection between ASM, effect on the immune system, and Th17.

According to the reviewer's comments, we have carefully reconsidered our manuscript.

In the **Extended Data Fig. 8a-g in the original manuscript**, we confirmed the effects of ASM immunization on Th17 cells and immune system in the AD mice. To more clearly show the therapeutic potential of ASM immunization in AD mice, we have moved these data to **Fig. 5e-l** in the revised manuscript.

Immunization with the ASM peptide in the APP/PS1 mice significantly reduced ASM activity and ceramide levels in the membrane of CD4⁺ T cells (**Fig. 5e in the revised manuscript**), resulting in a decrease of pathogenic Th17 cells and increase of Treg cells in the spleen and blood (**Fig. 5f,g in the revised manuscript**). Additionally, ASM immunization normalized the pro-inflammatory monocyte and macrophages in the blood of APP/PS1 mice, but not neutrophils (**Fig. 5h in the revised manuscript**). ASM-immunized APP/PS1 mice also showed reduction of pathogenic Th17 cells, monocytes, and even neutrophils in the brain, and increase of Treg cells and M2-like macrophages compared to vehicle-treated APP/PS1 mice (**Fig. 5i-l in the revised manuscript**). These data suggested that inhibition of plasma ASM activity following immunization of the APP/PS1 mice with an ASM peptide reduced pathogenic Th17 cells in the spleen, blood, and brain, and this was accompanied by normalization of other immune cell populations altered in these mice.

Fig. 5. Immunization with an ASM peptide prevents BBB disruption and memory impairment in APP/PS1 mice. **a**, Scheme of experimental procedures. **b**, Anti-ASM antibody titers (n = 7-8 per each group) and ASM activity (n = 8 per each group) in plasma. **c**, Ceramide concentration in plasma (n = 5 per each group). **d**, ASM activity in cortex and hippocampus (n = 4 per each group). **e**, ASM activity and ceramide levels in membrane of CD4⁺ T cell isolated from each mouse (n = 5 per each group). **f**, Percentage of Th17 cells in the spleen and blood (n = 4-5 per each group) and protein levels of IL17 in plasma (n = 4 per each group). **g**, Percentage of Treg cells in the spleen and blood (n = 5 per each group). **h**, Percentage of pro (Ly6c^{hi})- or anti (Ly6c^{low})-monocytes, macrophages, and neutrophils in the blood (n = 4 per each group). **i**, Percentage of Th17 cells in the brain (n = 4 per each group), protein levels of IL17 and mRNA levels of *Il17a* and *Il17f* (n = 4 per each group). **j**, Immunofluorescence images and quantification of Th17 cells (RORγ, red) in the perivascular (blood vessel, FITC-labeled albumin, green) and parenchyma cortex and hippocampus regions (n = 6 per each group). Scale bars, 20 µm. White arrow: perivascular Th17 cells. Extravascular cells were counted as parenchymal Th17 cells. **k**, Percentage of Treg cells in the brain (n = 5-6 per each group). **l**,

Percentage of monocytes, M1 or M2 macrophages, and neutrophils in the brain (n = 4-5 per each group). All immune cells were analyzed by flow cytometry. **m**, Representative *in vivo* time-lapse multiphoton imaging and quantification data of tetramethylrhodamine (TMR) dextran (MW = 40 kD; red) leakage from cortical vessels (n = 5 per each group). Scale bars, 50 μ m. **n**, Left, representative immunofluorescence images of thioflavin S (ThioS, A β plaques) in cortex and hippocampus. Scale bars, 50 μ m. Right, quantification of area occupied by A β plaques (n = 5-6 per each group). **o**, Learning and memory were assessed by the Morris water maze test (n = 14-20 mice per group). **p and q**, At probe trial day 11, time spent in target platform (p) and the number of times each animal entered the small target zone (q) were analyzed (n = 14-20 mice per group). **r**, The results of contextual and tone tasks during fear conditioning test (n = 12~17 mice per group). b-m,q, and r, One-way analysis of variance, Tukey's post hoc test. n and p, Student's t test. *P < 0.05, **P < 0.01, ***P < 0.001, ****P < 0.0001. The data were collected from 3 independent experiments. All error bars indicate s.e.m. All data analysis was done at 9-mo-old mice.

2. Immunization of wild type mice with ASM is missing.

We thank the reviewer for pointing this out. As recommended, we performed all experiments to include an ASM immunized-wide type (WT) group as shown in **Fig. 5 and Extended Data Fig. 9,10 of the revised manuscript**. ASM immunization of WT mice led to high ASM antibody titers and inhibition of ASM activity in the plasma compared to vehicle-treated WT mice. However, plasma ceramide levels and ASM activity in the cortex and hippocampus of WT mice were not altered by immunization with the ASM peptide. Moreover, ASM immunization of WT mice did not result in statistically significant alterations of immune cell populations, BBB damage, neuroinflammation, synapse loss, and learning and memory function.

In the revised manuscript, we have included the results regarding the ASM immunized-WT group in **Fig. 5 and Extended Data Fig. 9,10** with data analyses.

Fig. 5. Immunization with an ASM peptide prevents BBB disruption and memory impairment in APP/PS1 mice. **a**, Scheme of experimental procedures. **b**, Anti-ASM antibody titers ($n = 7-8$ per each group) and ASM activity ($n = 8$ per each group) in plasma. **c**, Ceramide concentration in plasma ($n = 5$ per each group). **d**, ASM activity in cortex and hippocampus ($n = 4$ per each group). **e**, ASM activity and ceramide levels in membrane of CD4⁺ T cell isolated from each mouse ($n = 5$ per each group). **f**, Percentage of Th17 cells in the spleen and blood ($n = 4-5$ per each group) and protein levels of IL17 in plasma ($n = 4$ per each group). **g**, Percentage of Treg cells in the spleen and blood ($n = 5$ per each group). **h**, Percentage of pro (Ly6c^{hi})- or anti (Ly6c^{low})-monocytes, macrophages, and neutrophils in the blood ($n = 4$ per each group). **i**, Percentage of Th17 cells in the brain ($n = 4$ per each group), protein levels of IL17 and mRNA levels of *Il17a* and *Il17f* ($n = 4$ per each group). **j**, Immunofluorescence images and quantification of Th17 cells (RORγ, red) in the perivascular (blood vessel, FITC-labeled albumin, green) and parenchyma cortex and hippocampus regions ($n = 6$ per each group). Scale bars, 20 µm. White arrow: perivascular Th17 cells. Extravascular cells were counted as parenchymal Th17 cells. **k**, Percentage of Treg cells in the brain ($n = 5-6$ per each group). **l**,

Percentage of monocytes, M1 or M2 macrophages, and neutrophils in the brain (n = 4-5 per each group). All immune cells were analyzed by flow cytometry. **m**, Representative *in vivo* time-lapse multiphoton imaging and quantification data of tetramethylrhodamine (TMR) dextran (MW = 40 kD; red) leakage from cortical vessels (n = 5 per each group). Scale bars, 50 μ m. **n**, Left, representative immunofluorescence images of thioflavin S (ThioS, A β plaques) in cortex and hippocampus. Scale bars, 50 μ m. Right, quantification of area occupied by A β plaques (n = 5-6 per each group). **o**, Learning and memory were assessed by the Morris water maze test (n = 14-20 mice per group). **p and q**, At probe trial day 11, time spent in target platform (p) and the number of times each animal entered the small target zone (q) were analyzed (n = 14-20 mice per group). **r**, The results of contextual and tone tasks during fear conditioning test (n = 12~17 mice per group). b-m,q, and r, One-way analysis of variance, Tukey's post hoc test. n and p, Student's t test. *P < 0.05, **P < 0.01, ***P < 0.001, ****P < 0.0001. The data were collected from 3 independent experiments. All error bars indicate s.e.m. All data analysis was done at 9-mo-old mice.

Extended Data Fig. 9. Immunization with an ASM peptide protects BBB disruption in APP/PS1 mice. **a**, Western blot analysis and quantification of tight junction proteins (Zo1, Claudin5, Occludin) in the cortex and hippocampus (n = 4 per each group). **b**, mRNA levels of tight junction (n = 4 per each group). **c**, Western blot analysis and quantification of fibrin and thrombin (n = 4 per each group). **d**, Top, immunofluorescence images of extravascular fibrin deposits in cortex and hippocampus. Scale bars, 50 μ m. Bottom, quantification of extravascular fibrin deposits (n = 6 per each group). **e**, Representative images and quantification of Lectin-positive microvascular profiles in cortex and hippocampus (n = 6 per each group). Scale bars, 50 μ m. **f**, Representative image and quantification of CD31, collagen IV, and aquaporin-4 in cortex and hippocampus (n = 6 per group). Scale bar: 50 μ m. **a-f**, One-way analysis of variance, Tukey's post hoc test. *P < 0.05, **P < 0.01, ***P < 0.001, ****P < 0.0001. The data were collected from 3 independent experiments. All error bars indicate s.e.m. All data analysis was done at 9-mo-old mice.

Extended Data Fig. 10. Immunization with an ASM peptide prevents neuroinflammation, impairment of microglia phagocytic function, A β accumulation, and synapse loss in APP/PS1 mice. **a**, Quantification of microglia (Iba-1) and astrocyte (GFAP) (n = 5 per each group). **b**, mRNA levels of inflammatory markers in the cortex and hippocampus (n = 4 per each group). **c**, Immunostaining images of the colocalization of microglia (Iba1, red) with A β aggregates (ThioS, green) and quantification of A β positive cells and microglia. (n = 5 mice per group). Scale bars = 10 μ m. **d**, Imaris-based automated quantification of microglial morphology (n = 4 per each group). **e**, Immunofluorescence images of ThioS (A β plaques, green) encapsulated within Lamp1+ structures (phagolysosomes, blue) in microglia (Iba1, red) present in cortex and hippocampus of each group. Scale bars, 20 μ m; 3D reconstruction from confocal image stacks scale bars, 10 μ m. Quantification of microglia volume occupied by Lamp1+ phagolysosomes, percent of microglia containing A β -loaded phagolysosomes and A β encapsulated in phagolysosomes (n = 4 mice per group). **f**, Quantification of 6E10 (n = 5 per each group). **g**, Analysis of A β 40 and A β 42 depositions using ELISA kits (n = 4 per each group). **h**, Representative images and quantification of amyloid angiopathy (n = 5 mice per group, CAA). To confirm amyloid angiopathy, brain section was stained with A β plaque (ThioS, green) and smooth muscle actin for vascular smooth muscle cells (SMA, red). Scale bars, 20

μ m. **i**, Western blot analysis for synaptophysin, synapsin1, PSD95, and MAP2 in each group (n = 4 mice per group). **j**, Representative swimming paths at day 10 of training of morris water maze test. a,b,i, One-way analysis of variance, Tukey's post hoc test. c-h, Student t-test. *P < 0.05, **P < 0.01, ***P < 0.001, ****P < 0.0001. The data were collected from 3 independent experiments. All error bars indicate s.e.m. All data analysis was done at 9-mo-old mice.

3. Morris Water Maze, the learning curve of the WT animals is very poor, although the overall differences after long period are significant.

According to the reviewer's comments, we have carefully reconsidered our data. **Fig. 5o** of the revised manuscript (**Fig. 6i in the original manuscript**) showed that the escape latency time of the WT-vehicle group was around 40s at Day 1 and the escape latency time was shorter by the day, indicating normal learning and memory function. Based on previous data using the same experimental Morris water maze equipment (**Additional Fig. 4**), we think that the data of the WT group in **Fig. 5o** are reproducible and accurate. To clear visualize the data, we have modified the x-axis of the graph in the **Fig. 5o** in the revised manuscript.

Fig. 5o. Learning and memory were assessed by the Morris water maze test (n = 14-20 mice per group). One-way analysis of variance, Tukey's post hoc test. **P < 0.01, ****P < 0.0001. The data were collected from 3 independent experiments. All error bars indicate s.e.m. All data analysis was done at 9-mo-old mice.

[FIGURE REDACTED]

Additional Fig. 4. [REDACTED]

Reviewer #2 (Remarks to the Author):

In this study Choi et al. investigate the molecular and cellular mechanisms downstream of elevated plasma ASM in an APP/PS1 mouse model of Alzheimer's disease. This builds on prior work from the same group on the role of ASM in brain aging and neurodegeneration. The authors describe an increase in plasma ASM in human MCI/AD and in 9 month old, but not 3 or 6 month old, APP/PS1 compared to control mice. To selectively investigate the role of systemic ASM, the authors combine a parabiosis approach with a transgenic mouse model overexpressing ASM in endothelial cells (ASM OE) leading to increased plasma ASM level. When comparing young APP mice parabiosed to ASM OE mice to those attached to WT mice, Choi et al. observe accelerated neurodegenerative phenotypes including increased Amyloid-beta pathology and microglial/astrocyte activation. Using a series of in vitro and in vivo experiments, the authors demonstrate that elevated ASM levels lead to several changes in immune cell populations including an increase in TH17 cells and a decrease in Tregs. These changes were observed in both peripherally in the blood and spleen as well as centrally in the brain. The authors go on to show that anti-IL17 treatment, which ablates TH17 cells, reduces pathological phenotypes in APP/PS1 mice parabiosed to ASM-OE mice. Lastly, using a series of active and passive anti-ASM immunization approaches, the authors show that blocking systemic ASM in both the ASM-OE conditions, and more importantly also in normal 7-9 months old APP/PS1 mice, reduces peripheral and central immune cell shifts, as well as Amyloid-beta pathology, glial activation, synaptic changes and behavioral impairments. This is a timely and compelling study on the systemic plasma factor ASM and its role on immune cell function, blood-brain barrier integrity and neurodegenerative changes in the brain. While the findings are of high quality and relevance, the experimental approaches are, at times, unnecessarily complicated and some of the data and conclusions can also be interpreted differently. Below, is a series of questions for the authors to address.

.

Major

1. It is somewhat surprising to see which experiments/data Choi et al. decided to feature in the main figures versus the supplement. At times, the reader is almost obliged to go through the extended figures, since they feature crucial data and experiments. For example, Figure 5 features a convoluted parabiosis experiment, in which a *Smpd1* knockout mouse is used to generate anti-bodies against ASM to block plasma ASM in its surgically connected APP/PS parabiotic partner. In Figure 6, the authors then go on use a much more controlled active immunization paradigm, that comes to the same conclusions, raising the question whether the data in Figure 5 is necessary at all. Conversely, in Extended Figure 10, the authors use passive immunization with a newly developed neutralizing antibody, which is arguably a much more therapeutically relevant approach. Importantly, both Figure 6 and Extended Figure 10 include WT baseline conditions which allow for a better assessment of the extent of the rescue, while Figure 5 does not. Moreover, full body *Smpd1* knockout mice in Figure 5 display additional changes such as large difference in neutrophils (Figure 5j,o), which further confounds the interpretation of this experiment. Given these confounds and limitations, it is unclear to us why Figure 5 was included in this manuscript. Correspondingly, we recommend the authors feature the Extended Figure 10 in the main manuscript in lieu of Figure 5.

We would like to thank the reviewer for these comments, and have carefully reconsidered our manuscript. As suggested by reviewer, we have moved original Fig. 5 to Extended Data Fig. 5 and moved the original Extended Data Fig. 10 to Fig. 6 in the revised manuscript.

2. The authors show that ASM is elevated in 9 but not 6 months old APP/PS1 mice (Extended Figure 1c). Similarly, the authors show that TH17 cells are increase in 9 months old APP/PS1 mice, although it is unclear if these changes are present at 3 or 6 months (Extended Figure 8b-g). While the authors used anti IL17 treatment in the accelerated ASM-OE parabiosis condition (Figure 3+4), it would be more informative to assess the benefits of this treatment in 6-9 months old APP/PS1 in the absence of ASM overexpression.

We thank the reviewer for pointing this out. As suggested, we confirmed whether Th17 cells are changed in samples of 3 or 6-month-old APP/PS1 mice in our system. The data showed no significant differences in Th17 cells in the spleen, blood, and brain of APP/PS1 mice at 3 or 6 months of age compared to WT mice, while 9-month-old APP/PS1 mice exhibited increase of

Th17 cells in each sample (**Additional Fig. 5**). These data are in agreement with an increase of plasma ASM activity in APP/PS1 mice at 9 months of age. Based on our data showing the relevance between plasma ASM and Th17 cells changes in APP/PS1 mice with age (**Extended Data Fig. 1c and Additional Fig. 5**), it is reasonable that Th17 cells were likely not to be increased in the absence of ASM overexpression in APP/PS1 mice. This is supported by data of Fig. 2 showing that ASM induces Th17 cell differentiation from CD4⁺ T cells. Also, in response to a question from Reviewer 1, we confirmed the effects of the IL17 antibody in 3-month-old APP/PS1 mice with no ASM activity (**Fig. 3a in the revised manuscript**). Although IL17 antibody injection induced reduction of Th17 cells in the spleen and blood of young APP/PS1 mice compared to a control antibody injected group, A β accumulation was not changed between the groups (**Fig. 3b,c in the revised manuscript**). Thus, IL17 antibody treatment would have no significant therapeutic effects in APP/PS1 mice in the absence of ASM overexpression.

To additionally examine the therapeutic effects of IL17 antibody in APP/PS1 mice exhibiting increased plasma ASM activity, we injected IL17 antibody (50 μ g/mouse, i.p. every other day) in 7.5-month-old APP/PS1 mice because ASM activity increases after 6 months of age in APP/PS1 mice (**Fig. 5a,b in the revised manuscript**). After 4 weeks of antibody injection, the morris water maze test was performed and then mice were sampled at 9 months of age (**Additional Fig. 6a**). Th17 cells were decreased in the spleen, blood, and brain of mice treated with the IL17 antibody compared to isotype control antibody (**Additional Fig. 6b**). Moreover, these mice showed decrease of A β accumulation and activation of microglia or astrocytes both in the cortex and hippocampus, contributing to improvement of memory dysfunction (**Additional Fig. 6c-h**). Therefore, these results indicated that inhibition of Th17 cells could prevent A β accumulation, neuroinflammation, and memory impairment in 9-month-old APP/PS1 mice. Th17 cell inhibition using IL17 antibody may therefore be an effective therapeutic approach for AD, but we believe that prophylactic effects of immunotherapy targeting plasma ASM is a more beneficial therapeutic approach because differentiation of Th17 cells is regulated by plasma ASM.

Additional Fig. 5. Percentage of Th17 cells through flow cytometry analysis in the spleen, blood, and brain of WT or APP/PS1 mice with age (n = 5 per group). Student's t test. **P < 0.01, ***P < 0.001. All error bars indicate s.e.m.

Additional Fig. 6. Treatment of IL17 antibody improves neuropathological features of APP/PS1 mice. **a**, Scheme of experimental procedures. IgG isotype control or IL17 antibody was injected (50 μ g/mouse, i.p. every other day) into 7.5-month-old of APP/PS1 mice for 5 weeks. **b**, Percentage of Th17 cells in the spleen, blood, and brain using flow cytometry (n = 5 per each group). **c**, Left, representative immunofluorescence images of thioflavin S (ThioS, A β plaques). Scale bars, 50 μ m. Right, quantification of area occupied by A β plaques (n = 5-7 per each group). **d** and **e**, Immunofluorescence images and quantification of microglia (h, Iba-1, red) and astrocyte (i, GFAP, red) with A β plaques (ThioS, green) (n = 5-6 per each group). Scale bars, 50 μ m. **f**, Learning and memory were assessed by the Morris water maze test (n = 10 per each group). **g** and **h**, At probe trial day 11, time spent in target platform (g) and the number of times each animal entered the small target zone (h) were analyzed (n = 10 per each group). b,d,e,f,h One-way analysis of variance, Tukey's post hoc test. c and g, Student's t test. *P < 0.05, **P < 0.01, ***P < 0.001, ****P < 0.0001. All error bars indicate s.e.m. All data analysis was done at 9-mo-old mice.

3. The authors lay out a narrative that ASM alters CD4 cell differentiation, which leads to an increase of TH17 cells. These peripheral TH17 cells then impair BBB function, migrate into the brain, where they drive neuroinflammatory changes, protein aggregation and functional impairments. While the authors provide some evidence supporting this narrative, a lot of their data can also have alternative interpretations and several aspects of this mechanism have not actually been definitively demonstrated. For example, while Choi et al show an increase in TH17 cells in the brain, they do not show that these cells are systemically derived from the blood. This could for example be shown using GFP labeled cells. Moreover, the increase/decrease in CNS TH17 cells is always paralleled by an increase/decrease in peripheral TH17 cells (Figure 3b,h +Extended Figure 3). Thus, the authors cannot experimentally distinguish between the effect of peripheral versus brain TH17 cells. While it is convenient to speculate that brain TH17 cells mediate all the effects, peripheral TH17 could also impair brain function by creating a more inflammatory systemic milieu. As such, several peripheral pro-inflammatory chemokines and cytokines have previously been shown to be able to enter the brain and negatively regulate brain function. While some of these questions may be challenging to experimentally address, the authors should at the least discuss these limitations and provide plausible alternative interpretations of their data in the discussion.

We thank the reviewer for these comments regarding the interpretation of our data.

According to the reviewer's comments, we performed additional experiment to confirm whether the Th17 cells increased in the brain of young APP/PS1 mice exposed to overexpressed plasma ASM were derived from blood using Il17a-EGFP knockin mice (stock number 018472, The Jackson Laboratory). This mouse possess an IRES-EGFP sequence after the stop codon of the Il17a gene, so that EGFP expression is limited to IL-17A-expressing cells. Splenic naive CD4⁺ T cells, which are EGFP⁻, were first purified from the EGFP mice using a CD4⁺ T cell isolation kit (Miltenyi biotec, 130-104-453), and then cultured with recombinant human ASM peptide (rASM, 2.5μM) under Th17 cell polarizing conditions (IL-6 (20 ng ml⁻¹), IL-23 (10 ng ml⁻¹), IL-1β (10 ng ml⁻¹), TGF-β1 (2 ng ml⁻¹), anti-IL-4 (10 ng ml⁻¹), anti-IFN-γ (10 μg ml⁻¹), and anti-IL-2 (10 μg ml⁻¹)) for 4 days to generate pathogenic EGFP⁺ Th17 cells.

Three-month-old APP/PS1 mice were then intraperitoneally injected with IgG isotype control or IL17 antibody (50 μg per mouse, every other day) during 5 weeks of exposure to the blood of ASM overexpressing Tie2-cre; *Smpd1*^{ox/ox} mice. One week before sampling, the culture-derived EGFP⁺ Th17 cells were intravenously injected (1×10⁶ cells per mouse) into

APP/PS1 mice in parabiosis with Tie2-cre; *Smpd1*^{ox/ox} mice that had been treated with IgG isotype control or IL17 antibody. Brain sections of each mouse was cut on a vibratome (30 μ m), and then stained with GFP (rat, 1:1000, Abcam, ab13970) and collagen IV (rabbit, 1:100, Abcam, ab6586) antibodies for quantification of perivascular and parenchymal EGFP⁺ Th17 cells. The sections were analyzed with a laser-scanning confocal microscope (FV3000; Olympus). Cell counts were obtained using MetaMorph software (Molecular Devices). Cell numbers per brain region were divided by the respective tissue area and represented as cells/mm².

Blood-derived EGFP⁺ Th17 cells were detected at high levels both in perivascular and parenchyma regions of the cortex and hippocampus of APP/PS1 mice exposed to blood of Tie2-cre; *Smpd1*^{ox/ox} mice treated with IgG isotype control antibody compared to mice exposed to blood of WT mice. IL17 antibody treatment led to reduction of these cells in each region of the brain (**Fig. 3k in the revised manuscript**). This data indicated that blood-derived pathogenic Th17 cells contributed to acceleration of neuropathological features in the brain of young APP/PS1 mice exposed to overexpressed plasma ASM. We have added this data as **Fig. 3k** in the revised manuscript.

As stated by the reviewer, we agree that cytokines and chemokines released from peripheral Th17 cells also could negatively affect brain function. We have described additional interpretations of our data in the discussion section of the revised manuscript according to reviewer's suggestion (p.25).

Fig. 3k. The representative images and quantification of blood-derived EGFP⁺-Th17 cells (green) in the perivascular (blood vessel, red, collagen IV) and parenchyma cortex and hippocampus regions of APP/PS1 mice exposed to blood of WT or Tie2-cre; *Smpd1*^{ox/ox} mice

with or without IL17 antibody (n = 6 per each group). Scale bar: 20µm. White arrow: perivascular EGFP⁺ Th17 cells. Extravascular cells were counted as parenchymal Th17 cells. AD, APP/PS1; OX, *Smpd1*^{ox/ox} (Tie2-cre; *Smpd1*^{ox/ox} mice). One-way analysis of variance, Tukey's post hoc test. *P < 0.05, **P < 0.01, ***P < 0.001, ****P < 0.0001. All error bars indicate s.e.m.

4. Immunostainings and quantifications. The authors should provide a more thorough characterization and description of several immunostainings and analyses. First, infiltrating brain immune cells were mostly analyzed by FACS of dissociated whole brain, thus obfuscating specific regional origins of the cells. While some immunostaining of cortex and hippocampus were performed, the authors should provide a more careful quantification of the infiltrating immune cells and distinguish between, vascular, perivascular and parenchymal localization. Additionally, Lectin labelling was predominantly used as a marker for the vasculature. However, Lectin does not necessarily label all blood vessels and additionally may also label reactive glial cells in AD mouse models. The authors should consider complementing their vascular analysis with additional vascular markers (e.g., CD31, CollagenIV or Aqp4).

Second, it is often unclear which exact cortical/hippocampal regions were quantified. The inclusion of a nuclear DAPI stain in representative images, a schematic depicting the quantified region or a more extensive description in the figure legend should be included. Additionally, several quantifications lack representative images. For example, in Extended Fig 7d/9i, the authors provide quantification of CAA. However, most mouse models of AD do not recapitulate CAA very robustly. Moreover, the authors do not provide any methodology of how CAA was assessed.

According to reviewer's comments, we have reconsidered and revised our manuscript.

Since it is difficult to get enough single cells for FACS analysis from the hippocampus, we initially characterized infiltrating immune cells using whole brains lacking cerebellum. Alternatively, we have now performed immunostaining of blood-derived pathogenic EGFP⁺ Th17 cells, which we propose is a major immune cell regulated by ASM, in the cortex and hippocampus, and these cells were carefully quantified by distinguishing between the perivascular and parenchyma regions. As shown in response to question 3, brain sections were stained with GFP (Abcam, ab13970) and collagen IV (Abcam, ab6586) antibodies for quantification of perivascular and parenchymal EGFP⁺ Th17 cells. The sections were analyzed with a laser-scanning confocal microscope (FV3000; Olympus). Cell counts were obtained

using MetaMorph software (Molecular Devices). The results showed that blood-derived pathogenic EGFP⁺ Th17 cells increased both in the perivascular and parenchyma regions of cortex and hippocampus of young APP/PS1 mice exposed to overexpressed plasma ASM. We have added these data as **Fig. 3k** and described the detailed experimental method in the revised manuscript (**p.38-39**). We also have added quantification of perivascular and parenchyma Th17 cells stained by Rory antibody in **Fig. 5j** and **Extended Data Fig. 5l** of the revised manuscript.

As suggested by the reviewer, we additionally performed vascular analysis with immunostaining using CD31 (goat, 1:100, R&D system, AF3628), Collagen IV (rabbit, 1:100, Abcam, ab6586), and Aquaporin-4 (chicken, 1:100, Synaptic Systems, 429 006) antibodies in the **Extended Data Fig. 6f** and **Extended Data Fig. 9f** of the revised manuscript. The results showed less damage of vascular structures stained with these antibodies in the cortex and hippocampus of APP/PS1 mice exposed to blood of *Smpd1*^{-/-} mice or with ASM immunization.

Extended Data Fig. 6f. The representative image and quantification of CD31, collagen IV, and aquaporin-4 in cortex and hippocampus of APP/PS1 mice exposed to blood of APP/PS1, WT, or *Smpd1*^{-/-} mice (n = 4 per each group). Scale bar: 50μm. One-way analysis of variance, Tukey's post hoc test. **P < 0.01, ***P < 0.001, ****P < 0.0001. All error bars indicate s.e.m. All data analysis was done at 9-mo-old mice.

Extended Data Fig. 9f. The representative image and quantification of CD31, collagen IV, and aquaporin-4 in cortex and hippocampus of WT and APP/PS1 mice with or without ASM

immunization (n = 6 per each group). Scale bar: 50 μ m. One-way analysis of variance, Tukey's post hoc test. **P < 0.01, ***P < 0.001, ****P < 0.0001. All error bars indicate s.e.m. All data analysis was done at 9-mo-old mice.

For quantification of immunostaining of cortex and hippocampus, four images in each region were obtained as shown in the schematic below (**Source Data 1 in the revised manuscript**), and then the intensity or cell count was quantified by MetaMorph software (Molecular Devices). We have added this schematic as **Source Data 1 in the revised manuscript** and described detailed quantification methods in the material and methods section of the revised manuscript (**p.38**). Furthermore, we have replaced some of the representative images including nuclear DAPI staining in the revised manuscript (**Fig. 1e,1h,1i, 5h, 6k** and **Extended Data Fig. 2b,2e,2f,8a**).

Source Data 1. Schematic depicting for quantification of immunostaining in cortex and hippocampus. Four images of 20x or 40x magnification within each region were obtained to quantification of immunostaining in this study. Red box: cortex region, Blue box: hippocampus region. Blue: DAPI, Green: ThioS.

To confirm amyloid angiopathy, brain sections were stained with Thioflavin S (Sigma-Aldrich, T1892) for A β and smooth muscle actin for vascular smooth muscle cells (SMA, Sigma-Aldrich, A2547) as described in previous studies (*J Exp Med.* 2014. 211:1551–1570; *Nature Communications.* 2018. 9:1479; *Nature Communications.* 2020. 11:2358; *PNAS*, 2022. 119:e2115082119). Four images were taken at 40x magnification within the cortex and hippocampus as shown Source Data 1. The percentage of A β intensity within each vessel stained with SMA antibody was measured by MetaMorph software (Molecular Devices). We have added representative images of CAA in the **Extended Data Fig. 8d and 10h** and described detailed quantification method in the material and methods section or Figure legends of the revised manuscript (**p.38**).

Extended Data Fig. 8d and 10h. The representative image and quantification of CAA in cortex and hippocampus of each mouse (n = 4 per each group). Brain section was stained with A β plaque (ThioS, green) and smooth muscle actin for vascular smooth muscle cells (SMA, red). 8d, One-way analysis of variance, Tukey's post hoc test. 10h, Student's t test. *P < 0.05, **P < 0.01. All error bars indicate s.e.m. All data analysis was done at 9-mo-old mice.

Minor

1. The authors should specify the mouse ages more clearly in the figure legends. This especially applies to the extended figures which, unlike the main figures, do not feature schematic with the experimental paradigms.

As suggested by the reviewer, we have added mouse ages in the Figure legends, especially in the legends of **Extended Data Fig. 2-4 of the revised manuscript**.

2. Figure 2. At what age was the serum collected that was used in the in vitro experiments? The authors should provide more experimental detail regarding plasma/serum collection methods including centrifugation speeds etc. Moreover, are the CD4 T cells derived from WT or from APP transgenic mice and at what age were they collected? Additionally, the authors appear to use an immortalized microglial cell line, BV2, when referring to the microglia experiments in vitro. However, BV2 cells are not microglia, which should be stated more clearly in the manuscript.

According to reviewer's comments, we have reconsidered and revised our manuscript.

In **Fig. 2a-f**, serum was collected from 3-month-old WT, *Smpd1*^{ox/ox}, and *Smpd1*^{-/-} mice. To collect serum, mouse blood was collected in a e-tube and the blood was allowed to clot by leaving it undisturbed at room temperature for 30 min. Serum was collected by centrifugation

(13,000 rpm, 4°C, 5min) and 5 % of serum was treated into the CD4⁺ T cells for *in vitro* experiments. We have added the age of mice in the Fig. 2 legend and detailed methods in the material and methods section of the revised manuscript (p.30).

In **Fig. 2g**, 3-month-old APP/PS1 mice and age- and weight-matched WT or *Smpd1*^{ox/ox} mice were selected for parabiosis surgery. After 5 weeks of parabiosis surgery, CD4⁺ T cells was isolated from the spleen of APP/PS1 mice parabiosis with WT or *Smpd1*^{ox/ox} mice, and membrane and cytosol protein were extracted for ASM activity and ceramide analysis. We have added the age of mice in the Fig. 2 legend of the revised manuscript.

The BV2 cells used in **Fig. 4a-c (Fig. 3l-n in the original manuscript)** are a type of microglial cell derived from C57/BL6 murine. This cell line is a well-characterized, extensively employed model system for microglia. As suggested by the reviewer, we have clearly stated and discussed this cell line in the revised manuscript (p.14).

3. What ASM peptide was used for the active immunization?

We used recombinant human ASM (rASM) peptide for active immunization in APP/PS1 mice. The synthesis of rASM was performed by Genscript. Briefly, Expi293F cells were transfected with the full-length human ASM cDNA plasmid. The cell culture supernatants were collected on day 6 were used for purification. Cell culture broth was centrifuged. Cell culture supernatant was loaded onto an affinity purification column at an appropriate flowrate. After washing and elution with appropriate buffers, the eluted fractions were pooled and buffer exchanged to the final formulation buffer. The purified protein was analyzed by SDS-PAGE, Western blot analysis to determine the molecular weight (68kDa) and purity. We have added detailed information of rASM used for active immunization in the material and methods section of the revised manuscript (p.43).

4. Several Western blots have very high background and it is unclear how the authors were able to quantify individual bands (eg Figure 4h)

According to the reviewer's comments, we have carefully reconsidered our western blot data. We re-performed western blots in **Fig. 4j and Extended Data Fig. 4g,6c,9c** and changed these images to clarify our results.

5. Figure 6 f: Representative in vivo time-lapse image versus the quantification show a 30min versus a 20min time-point.

According to the reviewer's comment, we have corrected this in **Fig. 5m (Fig. 6f in the original manuscript)** of the revised manuscript.

6. Do the authors have a possible explanation for the massive increase in peripheral neutrophils in the periphery (Figure 5j) and decrease in brain neutrophils (Figure 5o) in response to Smpd1 KO parabiosis?

According to the reviewer's comments, we have carefully reconsidered our manuscript. Neutrophils are the first line of innate immune defense against invading pathogens or acute infections. Thus, the increase of neutrophils in the blood is assumed to be the result of recognizing the ASM antibody as a foreign substance in the blood of APP/PS1 mice exposed to blood of *Smpd1*^{-/-} mice. Despite the neutrophil increase in the blood of these mice, the decrease of neutrophils in the brain might be due to decrease of pathogenic Th17 cells. ASM antibody in the blood of these mice inhibited ASM activity, leading to decrease of pathogenic Th17 cells both in blood and brain (**Extended Data Fig. 5b-h,k,l in the revised manuscript**). Since the cytokines released from Th17 cells such as IL17 are potent stimulators of neutrophil migration and infiltration (*Nat. Rev. Drug Discov.* 2012. 11:763-776; *Am. J. Pathol.* 2012. 181:8-18; *Prog. Neuropsychopharmacol. Biol. Psychiatry.* 2017. 79:408-419), reduction of pathogenic Th17 cell could impact on the decrease of neutrophil infiltration into the brain in these mice. Although neutrophils were increased in the blood of *Smpd1*^{-/-} blood-exposed APP/PS1 mice, the correction of other immune cells in the blood and brain might contribute to prevention of neuropathological changes in these mice as well.

Reviewer #3 (Remarks to the Author):

In this study, Choi et al described the pathogenic Th17 cell differentiation from CD4+ T cells and the acceleration of AD brain pathology and neurobiological behavioral impairments (i.e., spatial learning and memory decline in the Morris water maze) by increased ASM activity. This study also demonstrates the prevention of these phenotypes by *Smpd1* deletion and/or SAM-related active/passive immunotherapy. Overall, this is a very interesting and nicely conducted study that provides new insights into the cellular and molecular regulation of Th17 cell differentiation by elevated ASM and the consequent effects on AD pathogenesis. While I am enthusiastic about this novel and interesting study, there are a few areas in which this manuscript could be improved.

We would like to thank the reviewer for these comments.

Major comments:

1. Are all effects of ASM observed in this manuscript mediated by ceramides? While ASM activity in the plasma of AD patients and mouse models increases, no changes in the levels of plasma ceramides were detected (Extended Data Fig. 1a-d). More importantly, the APP/PS1-*Smpd1*^{ox/ox} parabiotic mice show increased plasma ASM activity compared to APP/PS1-WT parabiotic mice (Fig. 1b), whereas plasma ceramide levels are not significantly different in these mice (Fig. 1c). Clear discussion and explanations on how the “disassociation” between elevated plasma ASM activity and unchanged plasma ceramide level affects the observed protection against AD phenotypes in mice by the immunotherapy targeting plasma ASM will improve the manuscript.

According to the reviewer’s comments, we have carefully reconsidered our manuscript.

In **Fig. 1** of the revised manuscript, we observed that young APP/PS1 mice exposed to the blood of Tie2-cre; *Smpd1*^{ox/ox} mice showed an increase of plasma ASM activity, contributing to acceleration of A β accumulation and neuroinflammation, despite the fact that plasma ceramide levels are not changed. Additionally, we demonstrated the therapeutic potential of plasma ASM-targeting rather than plasma ceramide for AD treatment.

We also observed a significant increase of Th17 cells in the blood of young APP/PS1 mice exposed to blood of Tie2-cre; *Smpd1*^{ox/ox} mice (**Extended Data Fig. 3c in the revised manuscript**). To investigate whether increased plasma ASM mediates Th17 cell differentiation,

the serum of Tie2-cre; *Smpd1*^{ox/ox} mice was treated into the CD4⁺ T cells and then ASM activity was measured in cell membrane and cytosol fractions of the CD4⁺ T cells. CD4⁺ T cells treated with the serum of Tie2-cre; *Smpd1*^{ox/ox} mice showed an elevation of ASM activity in the cell membrane fraction, but not in cytosol (**Fig. 2e in the revised manuscript**). Increase of ASM activity in the CD4⁺ T cell membrane led to the increase of various ceramides in the cell membrane as well (**Fig. 2f in the revised manuscript**). We also found an increase of ASM activity and ceramide levels in the CD4⁺ T cell membranes derived from young APP/PS1 mice that had been exchanged with blood of Tie2-cre; *Smpd1*^{ox/ox} mice compared to WT mice (**Fig. 2g,h in the revised manuscript**). We proposed that the elevation of ceramide levels in the cell membrane was caused by the differentiation of pathogenic Th17 cells by stimulating downstream signaling related with Th17 cell differentiation (**Fig. 2j-l in the revised manuscript**). Collectively, these data indicated that ceramide in the membrane (not the plasma) of CD4⁺ T cells mediates pathogenic Th17 cell differentiation by elevation of plasma ASM, and contributes to acceleration of neuropathological features in young APP/PS1 mice. Also, it should be noted that ceramide is a particularly hydrophobic lipid that is not readily released from cells, and therefore changes in the membrane ceramides may not be reflected in the plasma.

Importantly, plasma ASM-targeting immunotherapy showed therapeutic effects on prevention neuropathological features in APP/PS1 mice. Although plasma ceramide levels were unchanged by ASM immunization as observed in the parabionts, ceramide levels in the CD4⁺ T cell membranes were decreased with the reduction of ASM activity, resulting in a decrease of pathogenic Th17 cells in the blood and brain and impacting on prevention of neuropathological changes in APP/PS1 mice. Therefore, these data suggest that plasma ASM-targeting immunotherapy could be a promising therapeutic strategy for AD.

To help the reviewer and reader better understand our study, we have added a detailed explanation of our results in the discussion section of the revised manuscript (**p.24**).

2. Fig. 2e, g, how does elevated plasma ASM in *Smpd1*^{ox/ox} serum lead to elevated ASM activity in CD4⁺ T cell membranes? Is the plasma ASM recruited to the cell membrane of CD4⁺ T cells or internalized and trafficked to vesicles or the lysosome of these cells? Is the membrane ASM of CD4⁺ T cells activated by some factors presented in *Smpd1*^{ox/ox} serum but absent in *Smpd1*^{-/-} serum? Are there other reasons? Please discuss and explain.

In **Fig. 2e,g** of the revised manuscript, we observed that CD4⁺ T cells treated with the serum of Tie2-cre; *Smpd1*^{ox/ox} mice showed an elevation of ASM activity in the cell membrane fraction, but not in cytosol. In vivo experiments also showed an increase of ASM activity only in the CD4⁺ T cell membranes derived from young APP/PS1 mice that had been exchanged with blood of Tie2-cre; *Smpd1*^{ox/ox} mice compared to WT mice. These results indicate that plasma ASM was recruited into the cell membranes rather than internalized into the cell.

To more clearly elucidate this, we performed additional experiments. CD4⁺ T cells were treated with the media containing His-tagged recombinant human ASM peptide (rASM, 1μM) and then protein was extracted from cell membrane and cytosol, respectively. The result of western blotting showed the presence of His-tagged rASM protein in cell membrane, but not in cytosol. Moreover, treatment of the media containing rASM and ASM antibody (23A12C3, 50μg/ml) reduced the expression of His-tagged rASM in the cell membrane (**Additional Fig. 7**). Furthermore, as shown in response to minor comment 2, rASM treatment induced an increase of ASM activity only in CD4⁺ T cell membrane, but not in cytosol (**Supplementary Fig. 1a in the revised manuscript**). Taken together, these data support the fact that plasma ASM is recruited into the cell membrane of CD4⁺ T cells.

Additional Fig. 7. Western blotting for His-tag ASM in the cytosol and membrane of CD4⁺ T cells after rASM with IgG isotype antibody and ASM antibody (23A12C3) treatment. GAPDH and caveolin-1 were used to confirm successful extraction of cytosol and membrane protein.

Supplementary Fig. 1. Recombinant human ASM induces CD4⁺ T cell apoptosis and pathogenic Th17 differentiation by increasing ceramide in cell membranes. **a**, ASM activity in membrane and cytosol of CD4⁺ T cells treated with or without rASM (n = 4 per each group). **b**, Ceramide concentration in membrane of CD4⁺ T cells (n = 4 per each group). **c**, Percentage of apoptotic cells detected with Annexin V⁺ in CD4⁺ T cells (n = 6 per each group). **d**, Representative flow cytometry plot and graph displaying the calculated percentage and mean fluorescent intensity of Th17 cells differentiated from CD4⁺ T cells (n = 6 per each group). **e**, Western blotting for p-stat3, p-JNK, p-AKT, and p-mTOR in Th17 cells differentiated from CD4⁺ T cell by rASM treatment (n = 4 per each group). a-e, Student's t test. **P < 0.01, ***P < 0.001, ****P < 0.0001. The data were collected from 3 independent experiments. All error bars indicate s.e.m.

3. Page 11, lines 201-204, “Collectively, these results (Fig. 2) suggested that elevated plasma ASM leads to elevated ASM activity and ceramide in CD4⁺ T cell membranes (Fig. 2e-h), resulting in apoptosis and pathogenic Th17 cell differentiation by stimulating downstream signals such as, Stat3, JNK, AKT, and mTOR (Fig. 2i-l).” Based on these data and this summary, is the conclusion that the membrane ASM and membrane ceramide are responsible for the observed effects of elevated ASM activity in AD patients and mouse models correct? If yes, the emphasize of the plasma ASM and the immunotherapy targeting plasma ASM in this manuscript might not be the best way to tone the study and the conclusion. If such a conclusion is not the conclusion of the authors, what conclusion should be drawn on this issue?

We thank the reviewer for pointing out this aspect. To help the reviewer better understand, we would like to explain as below.

Fig. 1 using the parabiosis model showed that elevation of plasma ASM activity led to acceleration of A β accumulation and neuroinflammation in the young APP/PS1 mice exchanged with blood of ASM overexpressing Tie2-cre; *Smpd1*^{ox/ox} mice, despite the fact that there was no change in brain ASM activity. In this regard, we observed a significant increase of Th17 cells in the blood of these mice, suggesting that plasma ASM is involved in Th17 cell differentiation from CD4⁺ T cells (**Extended Data Fig. 3c in the revised manuscript**).

The results of *in vitro* experiments indicated that CD4⁺ T cells treated with the serum of Tie2-cre; *Smpd1*^{ox/ox} mice showed an elevation of ASM activity in the cell membrane fraction, leading to an increase of ceramides in the cell membrane (**Fig. 2e,f in the revised manuscript**). We also found similar results in the CD4⁺ T cell membranes derived from young APP/PS1 mice that had been exchanged with blood of Tie2-cre; *Smpd1*^{ox/ox} mice compared to WT mice (**Fig. 2g,h in the revised manuscript**). The elevation of membrane ceramide levels mediated pathogenic Th17 cell differentiation (**Fig. 2j-l in the revised manuscript**). We also showed in the response to question 2 and minor comment 2 that exogenous rASM treated into CD4⁺ T cell media is recruited to the cell membranes. Treatment of the media containing rASM and ASM antibody (23A12C3) inhibited recruitment of rASM into the CD4⁺ T cell membranes (**Additional Fig. 7 and Supplementary Fig. 1 in the revised manuscript**). Overall, these results demonstrated that elevated plasma (exogenous) ASM is responsible for the increase of ASM and ceramide in the CD4⁺ T cell membrane, resulting in the increase of pathogenic Th17 cell differentiation and that it accelerates neuropathological features in young APP/PS1 mice.

Based on these results, antibody-based immunotherapy targeting plasma ASM showed

efficient inhibition of plasma ASM activity and CD4⁺ T cell membrane ASM activity and ceramide levels in AD mice, and this treatment prevented neuropathological features by reducing pathogenic Th17 cell (Fig. 5,6 and Extended Data Fig. 5-10 in the revised manuscript). These results support the importance of plasma ASM targeting for the prevention of AD pathologies.

As commented by the reviewer, we also think that CD4⁺ T cell membrane ASM/ceramide-targeting therapeutics could be beneficial for AD. However, we believe that plasma ASM-targeting immunotherapy could be a more promising and useful therapeutic approach for AD because elevated plasma ASM mediates the increase of CD4⁺ T cell membrane ASM/ceramide.

4. Indeed, it has been found that vesicles containing ASM can translocate to the plasma membrane and release ASM to the outer leaflet or to the extracellular space. This ASM species is referred to as secretory ASM (Rotolo et al., 2005), lysosomal ASM (Li et al., 2012; Defour et al., 2014), or ASM without specified affiliation (Grassmé et al., 2001; Verdurmen et al., 2010). Regardless, the term of “the plasma ASM” should be used to indicate the ASM in the fluid part of blood but it should not be also used for the membrane ASM of CD4⁺ T cells or other cells.

We thank the reviewer for pointing this out. As suggested, we have carefully reconsidered our manuscript and used the term “plasma ASM” to indicate ASM in the fluid part of blood in the revised manuscript.

Minor comments:

1. In Fig. 1i, the representative immunofluorescence images and quantification of astrocytes (i, GFAP) appear not to match. Why GFAP staining shows such a remarkable “region specific” in the cortex? If reactive astrocytes around A β plaques partially contribute to such an uneven distribution of astrocytes in the cortex brain region, why such a pattern is not seen in microglia distribution in the same brain region (h, Iba1)? Double-labelling of A β using 6E10 antibody and GFAP or Iba1 will be better.

According to the reviewer’s comments, we have carefully reconsidered our manuscript. Iba-1 is commonly used to stain both resident and activated microglia in the brain, while GFAP is

used to detect activation of astrocytes. For this reason, the pattern of Iba-1 and GFAP immunostaining may appear different in the AD brain. As commented by reviewer, we performed double-staining of A β and Iba-1 or GFAP in the cortex and hippocampus of 3-month-old of APP/PS1 mice exchanged by parabiosis with blood of WT or Tie2-cre; *Smpd1*^{ox/ox} mice. The data showed region-specific activation of microglia and astrocytes around A β plaques in the cortex and hippocampus of APP/PS1 mice exchanged with blood of Tie2-cre; *Smpd1*^{ox/ox} mice. We have changed these data in the revised manuscript (**Fig. 1h,i and Extended Data Fig. 2e,f**).

Fig. 1h,i. Representative images and quantification of Iba-1 (red) and GFAP (red) with A β (green, ThioS) in the cortex (n = 5-7 per group). Scale bars, 20 μ m. AD, APP/PS1; OX, *Smpd1*^{ox/ox} (Tie2-cre; *Smpd1*^{ox/ox} mice). One-way analysis of variance, Tukey's post hoc test. ***P < 0.001, ****P < 0.0001. All error bars indicate s.e.m. All data analysis was done at 4.5-mo-old mice.

Extended Data Fig. 2e,f. Representative images and quantification of Iba-1 (red) and GFAP (red) with A β (green) in the hippocampus. Scale bars, 20 μ m. AD, APP/PS1; OX, *Smpd1*^{ox/ox} (Tie2-cre; *Smpd1*^{ox/ox} mice). One-way analysis of variance, Tukey's post hoc test. ***P < 0.001, ****P < 0.0001. All error bars indicate s.e.m. All data analysis was done at 4.5-mo-old mice.

2. In Fig. 2l, does ASM treatment cause similar activation of these pathways as did by C-16 ceramide? Is there any possible involvement of the other hydrolysate of sphingomyelin, phosphorylcholine in the effects of elevated ASM activity in this manuscript?

According to the reviewer's comment, we performed additional experiments to investigate whether ASM causes apoptosis and differentiation of CD4⁺ T cells as did by C-16 ceramide. Recombinant human ASM peptide (rASM, 2.5μM) was treated into the CD4⁺ T cells, and then ASM activity was measured in cell membrane and cytosol fractions. The results showed that rASM treatment induced an increase of ASM activity in the cell membrane, but not in cytosol, leading to an increase of various kind of ceramides in the cell membrane (**Supplementary Fig. 1a,b**). rASM treatment also caused apoptosis and Th17 cell differentiation of CD4⁺ T cells (**Supplementary Fig. 1c,d**). Additionally, it induced phosphorylation of Th17 cell intracellular signals such as Stat3, JNK, AKT, and mTOR (**Supplementary Fig. 1e**). In **Fig. 6f** of the revised manuscript, we also observed that the serum from *Smpd1*^{ox/ox} mice exhibiting overexpression of plasma ASM led to phosphorylation of these signals.

As commented by the reviewer, ASM catalyzes the hydrolysis of sphingomyelin to ceramide and phosphorylcholine. To confirm the involvement of phosphorylcholine in Th17 cell differentiation, we treated phosphorylcholine (10μM, MedChemExpress, HY-B2233B) into the CD4⁺ T cells. The data exhibited no significant difference in Th17 cell differentiation (**Supplementary Fig. 2**). Therefore, these data indicated that rASM causes apoptosis and Th17 cell differentiation through ceramides in CD4⁺ T cell membranes.

We have added these data in the **Supplementary Fig. 1,2** of the revised manuscript.

Supplementary Fig. 1. Recombinant human ASM induces CD4⁺ T cell apoptosis and pathogenic Th17 differentiation by increasing ceramide in cell membranes. **a**, ASM activity in membrane and cytosol of CD4⁺ T cells treated with or without rASM (n = 4 per each group). **b**, Ceramide concentration in membrane of CD4⁺ T cells (n = 4 per each group). **c**, Percentage of apoptotic cells detected with Annexin V⁺ in CD4⁺ T cells (n = 6 per each group). **d**, Representative flow cytometry plot and graph displaying the calculated percentage and mean fluorescent intensity of Th17 cells differentiated from CD4⁺ T cells (n = 6 per each group). **e**, Western blotting for p-stat3, p-JNK, p-AKT, and p-mTOR in Th17 cells differentiated from CD4⁺ T cells by rASM treatment (n = 4 per each group). a-e, Student's t test. **P < 0.01, ***P < 0.001, ****P < 0.0001. The data were collected from 3 independent experiments. All error bars indicate s.e.m.

Supplementary Fig. 2. Phosphorylcholine does not induce Th17 cell differentiation from CD4⁺ T cells. Representative flow cytometry plot and graph displaying the calculated percentage and mean fluorescent intensity of Th17 cells differentiated from CD4⁺ T cells treated with or without phosphorylcholine (n = 6 per each group). Student's t test. The data were collected from 3 independent experiments. All error bars indicate s.e.m.

3. Multi-dimensional views of microglial phagocytosis of Fluor 555-labelled A β in Fig. 1i and Fig. 3m that include image information in the x, y, and z dimensions are better presentation of phagocytic function.

According to the reviewer's comment, we have included images with x, y, and z dimensions in the **Fig. 11** and **Fig. 4c** of the revised manuscript.

Fig. 11. Immunofluorescence images of ThioS (A β plaques, green) encapsulated within Lamp1⁺ structures (phagolysosomes, blue) in microglia (Iba1, red) present in cortex of APP/PS1 mouse in parabiosis with Smpd1ox/ox mice. Scale bars, 30 μ m.

Fig. 4c. Left, immunofluorescence images of BV2 microglial cells with Fluor 555-labeled A β in each group. Scale bars = 50 μ m. Right, quantification of the Fluor 555-labeled A β uptake (n = 6 per each group). One-way analysis of variance, Tukey's post hoc test. **P < 0.01, ****P < 0.0001. The data were collected from 3 independent experiments. All error bars indicate s.e.m.

4. In Fig. 3k-m, IgG isotype/vehicle control, as did in Fig. 3a-j, will be better than no treatment control.

We thank the reviewer for pointing this out. We missed the notation of IgG isotype control antibody in **Fig. 3l-n** (**Fig. 3k-m in the original manuscript**) and also other figures (**Fig. 6**), and have now added this notation.

5. Overall this manuscript is well-written. However, some result description should be more accurate. For examples, "Co-culture of microglia with the ASM treated Th17 cells revealed a

deficient Fluor 555-labeled A β phagocytic capacity of the microglia, while Th17 cells treated with ASM or IL17 antibody did not exhibit this abnormal microglia function (Fig. 3m). (lines 236 & 237). The inaccurate writing in the clause could cause confusion or misunderstanding. How Th17 cells could exhibit microglia function? Should the clause be “while microglia co-cultured with ASM-primed Th17 cells treated with IL17 antibody did not exhibit this abnormal function”?

We apologize for confusion or misunderstanding in the original manuscript, and have corrected this sentence in the revised manuscript (p.14).

6. Lines 346-349, “These data revealed the significant positive effects of immunization of APP/PS1 mice with an ASM peptide on various pathological features, including changes of immune cell populations, BBB damage, neuroinflammation, A β accumulation, synapse loss, and learning and memory impairment”. Lines 44-48, “Antibody-based immunotherapy targeting plasma ASM showed efficient inhibition of ASM activity in the blood of AD mice and, interestingly, this treatment led to positive effects on blood-brain barrier (BBB) disruption, neuroinflammation, A β deposits, synapse loss, and memory impairment by suppressing pathogenic Th17 cells”. My suggestion is to replace the “positive effects” in these two sentences with prophylactic effects to make them clearer.

According to the reviewer’s comment, we have revised this sentence in the revised manuscript (p.2, 20).

REVIEWER COMMENTS

Reviewer #1 (Remarks to the Author):

The authors have addressed most the reviewers' comments in a carefully way.

Reviewer #2 (Remarks to the Author):

The authors have provided an extensive amount of new data, and have satisfactorily addressed previously raised concerns.

Reviewer #3 (Remarks to the Author):

The authors have addressed my concerns except the minor comment 1.

Original minor comment 1:

In Fig. 1i, the representative immunofluorescence images and quantification of astrocytes (i, GFAP) appear not to match. Why GFAP staining shows such a remarkable "region specific" in the cortex? If reactive astrocytes around A β plaques partially contribute to such an uneven distribution of astrocytes in the cortex brain region, why such a pattern is not seen in microglia distribution in the same brain region (h, Iba1)? Double-labelling of A β using 6E10 antibody and GFAP or Iba1 will be better.

New comments:

In the response letter, the statement that "GFAP is used to detect activation of astrocytes" is incorrect. In revised manuscript, why Fig. 1l and Extended Data Fig. 2f show so dramatically different staining patterns of GFAP: GFAP positive staining in resting astrocytes in Extended Data Fig. 2f but not in Fig. 1l.

The statements "Iba-1 is commonly used to stain both resident and activated microglia in the brain, while GFAP is used to detect activation of astrocytes. For this reason, the pattern of Iba-1 and GFAP immunostaining may appear different in the AD brain."

The authors should use sister brain slices to do the immuno-staining in Fig. 1h, l. Similarly, sister brain slices should be used in Extended Data Fig. 2e, f. Simultaneously, the authors should also do double staining of Iba1 and GFAP. These data together can clearly show whether the pattern of Iba-1 and GFAP immunostaining is different in the AD brain in your manuscript. Based on "GFAP is used to detect activation of astrocytes" to claim "the pattern of Iba-1 and GFAP immunostaining may appear different in the AD brain" is not right.

Because the representative immunofluorescence images and quantification of astrocytes (i, GFAP burden) in revised Fig. 1l still appear not to match. The authors should include very detailed methods for quantification of GFAP burden in these figures.

"The data showed region-specific activation of microglia and astrocytes around A β plaques in the cortex and hippocampus of APP/PS1 mice exchanged with blood of Tie2-cre; Smpd1ox/ox mice". New data in the revised manuscript (Fig. 1h,i and Extended Data Fig. 2e,f) and original data cannot convincingly support such a conclusion. What do the author means "region-specific activation of microglia and astrocytes around A β plaques.....". The activation pattern of microglia or astrocytes is not only related to A β plaques.

Reviewer #3 (Remarks to the Author):

The authors have addressed my concerns except the minor comment 1.

Original minor comment 1:

In Fig. 1i, the representative immunofluorescence images and quantification of astrocytes (i, GFAP) appear not to match. Why GFAP staining shows such a remarkable “region specific” in the cortex? If reactive astrocytes around A β plaques partially contribute to such an uneven distribution of astrocytes in the cortex brain region, why such a pattern is not seen in microglia distribution in the same brain region (h, Iba1)? Double-labelling of A β using 6E10 antibody and GFAP or Iba1 will be better.

New comments:

In the response letter, the statement that “GFAP is used to detect activation of astrocytes” is incorrect. In revised manuscript, why Fig. 1I and Extended Data Fig. 2f show so dramatically different staining patterns of GFAP: GFAP positive staining in resting astrocytes in Extended Data Fig. 2f but not in Fig. 1I.

The statements “Iba-1 is commonly used to stain both resident and activated microglia in the brain, while GFAP is used to detect activation of astrocytes. For this reason, the pattern of Iba-1 and GFAP immunostaining may appear different in the AD brain.”

The authors should use sister brain slices to do the immuno-staining in Fig. 1h, I. Similarly, sister brain slices should be used in Extended Data Fig. 2e, f. Simultaneously, the authors should also do double staining of Iba1 and GFAP. These data together can clearly show whether the pattern of Iba-1 and GFAP immunostaining is different in the AD brain in your manuscript. Based on “GFAP is used to detect activation of astrocytes” to claim “the pattern of Iba-1 and GFAP immunostaining may appear different in the AD brain” is not right.

Because the representative immunofluorescence images and quantification of astrocytes (i, GFAP burden) in revised Fig. 1I still appear not to match. The authors should include very detailed methods for quantification of GFAP burden in these figures.

“The data showed region-specific activation of microglia and astrocytes around A β plaques in the cortex and hippocampus of APP/PS1 mice exchanged with blood of Tie2-cre; Smpd1ox/ox mice”. New data in the revised manuscript (Fig. 1h,i and Extended Data Fig. 2e,f) and original data cannot convincingly support such a conclusion. What do the author means “region-specific activation of microglia and astrocytes around A β plaques.....”. The activation pattern of microglia or astrocytes is not only related to A β plaques.

We apologize to the reviewer for any confusion and misunderstand due to the lack of detailed explanation about our data.

Previous work, including our own, showed that GFAP expression is not prominent in the cortex, while high in the hippocampus of wild type mouse (**Additional Figure 1**: *Glia*. 2016. 64:240-54; *Hypertension*. 2019. 73:217-228; *Mol Neurobiol*. 2020. 57:3727-3743; *Aging*. 2021. 13:6634-6661; *Science Advances*. 2021. 7:eabe3600; *Neurotherapeutics*. 2022. 19:1546-1565; *PNAS*. 2022. 119:e2115082119), indicating that the GFAP expression appears differently depending on the brain region in normal condition. Thus, the different staining patterns of GFAP in the cortex and hippocampus have been accepted as characteristics of the marker.

[FIGURE REDACTED]

Additional Figure 1. References of GFAP immunostaining in the cortex and hippocampus of wild type and APP/PS1 mice.

Since Iba-1 is a marker with characteristics of detecting both resident and activated microglia in the brain, it is evenly observed in the brain parenchyma and strongly detected around A β plaque in the cortex of AD mouse model. However, GFAP expression is prominently observed around the A β plaque in the cortex of AD mouse model. Many previous studies have shown different staining patterns of Iba-1 and GFAP in the AD cortex (**Additional Figure 2**: *PLoS One*. 2014. 9:e111215; *Mol Neurobiol*. 2020. 57:3727-3743; *Journal of Neuroinflammation*. 2020. 17:302; *Exp Neurol*. 2021. 336:113506; *Response data for reviewer 2 in this study*).

[FIGURE REDACTED]

Additional Figure 2. References of Iba-1 and GFAP immunostaining with A β plaque in the cortex and hippocampus of wild type and APP/PS1 mice.

The immunostaining for Iba-1 and GFAP in Fig. 1h,i and Supplementary Fig 2e,f (original Figure: Extended Data Fig 2e,f) were resulted from using sister brain slices. To confirm

different staining patterns of Iba-1 and GFAP, we performed double staining of Iba-1 and GFAP as suggested by reviewer. The images showed that Iba-1 was detected in the cortex and hippocampus of all group, and its expression was increased around A β plaque in the cortex and hippocampus of AD (APP/PS1) mouse exchanged with blood of OX (Tie2-cre; *Smpd1*^{ox/ox}) mouse. GFAP expression is observed in the hippocampus but relatively low in the cortex of all group except for AD mouse from AD-OX pair. However, its expression was increased around A β plaque in the cortex and hippocampus of AD mouse from AD-OX pair (**Additional Figure 3**). This result showed a slightly different pattern of Iba-1 and GFAP immunostaining in the brain of AD mouse from AD-OX pair.

Additional Figure 3. Representative images of Iba-1 (red) and GFAP (blue) double immunostaining with A β plaque (ThioS, green) in the cortex and hippocampus. Scale bar = 500 μ m.

For quantification of GFAP expression in the Fig. 1i, the four images were taken at 40x magnification within cortex, and the images were imported into the MetaMorph software. The “Set color threshold tool” and “Show region statistics tool” was used to quantify the intensity

of GFAP, according to the software manual. Although GFAP expression is not prominent in the images of the cortex except for AD mice in AD-OX pairs, fine expression can be detected by setting in the software. Therefore, we believe that quantitative value of our result is considered an accurate value because the software settings for quantification are applied to all images. We have added detailed methods for GFAP quantification in the revised manuscript and hope the reviewer understands for this technical situation.

Regarding the sentences pointed out by the reviewer, we apologize for the confusion caused by excessive interpretation of our data. We described about these data “these mice exhibited highly activated microglia and astrocyte (Fig. 1h,i and Supplementary Fig 2e,f (original Figure: Extended Data Fig 2e,f))” without the word “region-specific activation” in the revised manuscript.

We again apologize to the reviewer for any confusion about our data.